# LANGUAGE MODELS AS ZERO-SHOT PLANNERS: EXTRACTING ACTIONABLE KNOWLEDGE FOR EMBODIED AGENTS

## ABSTRACT

Can world knowledge learned by large language models (LLMs) be used to act in interactive environments? In this paper, we investigate the possibility of grounding high-level tasks, expressed in natural language (i.e. "make breakfast"), to a chosen set of actionable steps (i.e. "open fridge"). While prior work focused on learning from explicit step-by-step examples of how to act, we surprisingly find that if pre-trained LMs are large enough and prompted appropriately, they can effectively decompose high-level tasks into low-level plans without any further training. However, the plans produced naively by LLMs often cannot map precisely to admissible actions. We propose a procedure that conditions on existing demonstrations and semantically translates the plans to admissible actions. Our evaluation in the recent VirtualHome environment shows that the resulting method substantially improves executability over the LLM baseline. The conducted human evaluation reveals a trade-off between executability and correctness but shows a promising sign towards extracting actionable knowledge from language models[1].

## 1 INTRODUCTION

Large language models (LLMs) have made impressive advances in language generation and understanding in recent years (Devlin et al., 2018; Radford et al., 2019; Raffel et al., 2019; Brown et al., 2020). See Bommasani et al. (2021) for a recent summary of their capabilities and impacts. Being trained on large corpora of human-produced language, these models are thought to contain a lot of information about the world (Roberts et al., 2020; Li et al., 2021; BIG-bench collaboration, 2021) - albeit in linguistic form.

We ask whether we can use such knowledge contained in LLMs not just for linguistic tasks, but to make goal-driven decisions to that can be enacted in interactive, embodied environments. But we are not simply interested in whether we can train models on a dataset of demonstrations collected for some specific environment – we are instead interested in whether LLMs *already contain* information necessary to accomplish goals without any additional training.

More specifically, we ask whether world knowledge about how to perform high-level tasks (such as "make breakfast") can be expanded to a series of groundable actions (such as "open fridge", "grab milk", "close fridge", etc) that can be executed in the environment. For our investigation, we use recently proposed VirtualHome environment (Puig et al., 2018). It can simulate a large variety of realistic human activities in a household environment and supports ability to perform them via embodied actions defined with a `verb-object` syntax. However, due to open-ended nature of the tasks, it is difficult to autonomously evaluate their success. We rely on human evaluation (conducted on Mechanical Turk) to decide whether sequences of actions meaningfully accomplish posed tasks.

We find that large GPT-3 (Brown et al., 2020) and Codex (Chen et al., 2021) models, when prompted with a single fixed example of a task description and its associated sequence of actions, can produce very plausible action plans for the task we're interested in. Such completions reflect the information already stored in the model – no model fine-tuning is involved. Additionally, we only observe this

---

[1]Results and videos at `https://sites.google.com/view/language-model-as-planner`

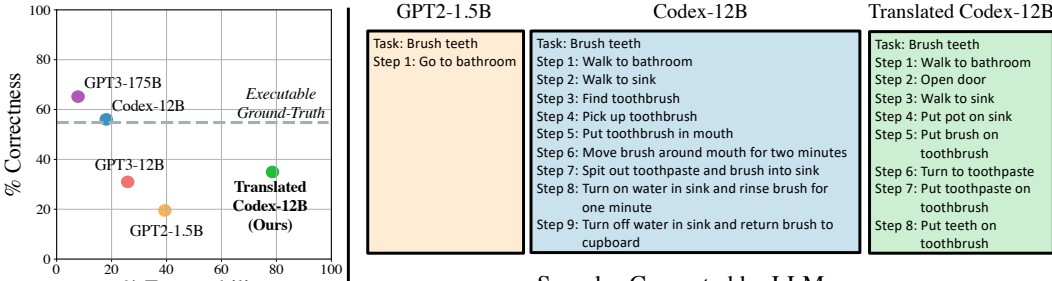

Samples Generated by LLMs

Figure 1: Executability v.s. semantic correctness of generated action plans (**left**) and sample action plans generated by different models (**right**). Large models can produce action plans indistinguishable from plans created by humans, but frequently are not executable in the environment. Using our techniques, we can significantly improve executability, albeit at the cost of correctness.

effect in the larger models. Unfortunately, despite their semantic correctness, the produced action plans are often not executable in the environment. Produced actions may not map precisely to admissible actions, or may contain various linguistic ambiguities.

We propose several tools to improve executability of the model's outputs. First, we enumerate all admissible action phrases and map the model's output action to the most semantically-similar admissible action (we use similarity measure between sentence embeddings produced by a RoBERTa model Liu et al. (2019) in this work, but other choices are possible). Second, we use the model to autoregressively generate actions in a plan by conditioning past actions that have been made admissible via the technique above. Such on the fly correction can keep generation anchored to admissible actions. Third, we provide weak supervision to the model by prompting the model with a known task example similar to the query task. This is somewhat reminiscent of prompt tuning approaches, but does not require access to gradients or internals of the model.

Using above tools to bias model generation, we find that we improve executability of instructions from 18% to 79% (see Figure 1) without any invasive modifications to model parameters or any extra gradient or internal information beyond what is returned from the model's forward pass. This is advantageous because it does not require any modifications to model training procedure and can fit within existing model serving pipelines. However, we do find there to be a significant drop in correctness of the instruction sequences generated with above tools (as judged by humans), indicating a promising step, but requiring more research on the topic.

To summarize, our paper's contributions are as follows:

- We show that without any training, large language models can be prompted to generate plausible goal-driven action plans, but such plans are frequently not executable in interactive environments.

- We propose several tools to improve executability of the model generation without invasive probing or modifications to the model.

- We conduct a human evaluation of multiple techniques and models and report on the trade-offs between executabiltiy and semantic correctness.

## 2    EVALUATION FRAMEWORK

Simulating open-ended tasks that resemble naturalistic human activities requires an environment to support a rich set of diverse interactions, rendering most existing embodied environments unsuitable for our investigation. One exception is VirtualHome (Puig et al., 2018), which models human activities in a typical household. Therefore, we only provide evaluation in this environment. To further measure correctness given open-ended tasks, we conduct a human evaluation. We note that since no further training is involved throughout our investigations, the observations and findings presented in this paper should also translate to similar embodied environments.

## 2.1 EVALUATED ENVIRONMENT: VIRTUALHOME

**Preliminaries**  In VirtualHome, activities are expressed as programs. Each program consists of a sequence of steps, where each step is written as: $[action]$ $\langle arg_1 \rangle (id_1)$ ... $\langle arg_n \rangle (id_n)$. Each $action$ refers to atomic actions such as "walk", "open", and "put". A total of 45 atomic actions are supported by VirtualHome. Different actions take in different numbers of $arg$ necessary for specifying an interaction. Associated with each $arg$ is a unique $id$ specifying the corresponding node in the environment graph, in case of multiple instances of the same object class are present in the graph. For the sake of simplicity, we omit the $id$ in the remaining discussions of this paper and allow automatic assignment by the environment. An example program is shown in Appendix 4.

**Evaluated Tasks**  We use the knowledge base collected by VirtualHome for evaluation. The knowledge base contains household activities crowd-sourced from Amazon Mechanical Turk (MTurk). The MTurk workers were asked to provide natural language descriptions of daily household activities and all actionable steps necessary for completing the activities. The descriptions are both given as high-level task descriptions and step-by-step instructions. We omit the use of step-by-step instructions in this work as we desire direct extraction of executable programs from only task descriptions. For evaluations, we randomly sample a subset of 88 high-level tasks, each having one or more annotated ground-truth programs. The remaining 204 tasks are used as *demonstration set*, from which we are allowed to select as example(s) for prompting language models. Note that no training or fine-tuning is performed using these tasks and their annotations. More details of the evaluated tasks can be found in Appendix 8.6.

## 2.2 METRICS

A program that commands the agent to wander around in a household environment is highly executable but may not complete the desired task. On the other hand, a program composed of step instructions from knowledge bases can likely complete the task but cannot be executed. The reason is that free-form instructions can be ambiguous and may lack necessary common-sense actions. To this end, we consider two axes for evaluation: **executability** and **correctness**.

**Executability**  Executability measures whether an action plan can be *correctly parsed* and *satisfies the common-sense constraints* of the environment. To be correctly parsed, an action plan must be syntatically correct and contain only allowed actions and recognizable objects. To satisfy the common-sense constraints, each action step must not violate the set of its pre-conditions (e.g. the agent cannot grab milk from the fridge before opening it) and post-conditions (e.g. the state of the fridge changes from "closed" to "open" after the agent opens it). We report the average executability across all 88 tasks and across all 7 VirtualHome scenes.

**Correctness**  Unlike most embodied environments where the completion of a task can be easily judged, the ambiguous and multimodal nature of natural language task specification makes it impractical to obtain a gold-standard measurement of correctness. One approach could be measuring similarity of the final environment state produced by executing predicted and ground-truth programs, but VirtualHome initializes an environment differently based on the to-be-executed program, making comparisons difficult if measured in such way. Therefore, we conduct human evaluation for the highlighted methods. More details of the human evaluations can be found in Appendix 8.5. For the remaining methods and ablation studies, we rely on a match-based metric that measures how similar a generated program is to human annotations. Specifically, we follow Puig et al. (2018) and calculate the longest common subsequence (LCS) between two programs, normalized by the maximum length of the two. In the presence of multiple ground-truth programs for a single task, we take the maximum LCS across the ground-truth programs. However, we note that the majority of the tasks only have one ground-truth annotation, but there are often many plausible ways to complete a certain task, making this metric imperfect at evaluation program correctness[2]. Although correlation between the two is shown by Puig et al. (2018), we consider it only as a proxy metric in replacement of unscalable human evaluation.

---

[2]Although LCS has a mathematical range of $[0, 1]$, we measure the LCS between different ground-truth programs for the same task and find an empirical maximum of $0.489$.

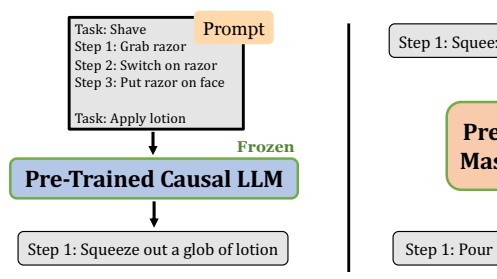 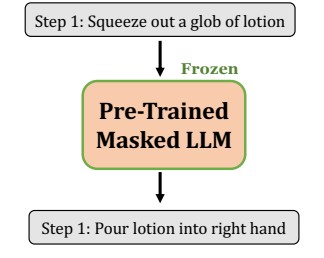 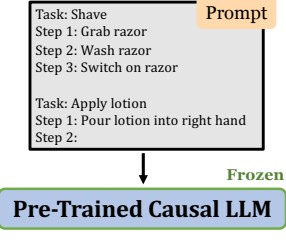

Zero-Shot Planning via Causal LLM   Translation to Admissible Action   Step-By-Step Autoregressive Generation

Figure 2: We investigate the possibility of extracting actionable knowledge from pre-trained language models without any additional training. We first show surprising finding that large language models (LLMs) can decompose high-level tasks into sensible low-level action plans (left). To make the action plans executable, we propose to translate each step into admissible action via another LLM (middle). The translated action is appended to the original prompt used for generating the remaining steps (right).

## 3 METHOD

In this section, we investigate the possibility of extracting actionable knowledge from pre-trained language models without further training. We first give an overview of the common approach to query large language models (LLMs) and how it may be used for embodied agents. Then we describe an inference-time procedure that addresses several deficiencies of the LLM baseline and offers better executability in embodied environments. We break down the proposed procedure into three individual components, each discussed in Sec 3.2, 3.3, and 3.4.

Since LMs excel at dealing with natural language text instead of the specific format required by VirtualHome as described in Section 2.1, we only expose natural language text to LMs. To do this, we define a mapping for each atomic action that parses a natural language phrase to the required format. For instance, "Walk to living room" is converted to "[Walk] ⟨living_room⟩(1)". When an LM output cannot be parsed to any of the allowed action, the entire program is considered syntactically incorrect and thus not executable.

### 3.1 PROMPTING

Previous works have shown that large language models pre-trained on a colossal amount of data contain useful world knowledge that can be probed to perform various down-stream tasks (Radford et al., 2019; Brown et al., 2020). Notably, autoregressive LLMs can even perform in-context learning, an approach to solve tasks using only contextual information without gradient updates (Brown et al., 2020). Contextual information is given as part of the input prompt and LMs are asked to complete the remaining text. It often consists of natural language instructions and/or a number of examples containing the desired input/output pairs.

We adopt the same approach to query LLMs to generate action plans for high-level tasks. Specifically, we prepend one example task description sentence and its annotated action plan from the *demonstration set* to the query task description, as shown in Fig 2. To obtain text completion results, we sample from autoregressive LLM using temperature sampling and nucleus sampling (Holtzman et al., 2019). We refer to this LM as **Planning LM** and the approach using this LM for plan generation as **Vanilla *[LM]***, where *[LM]* is replaced by specific language model such as GPT-3 or Codex.

To further improve the quality of the generated output, we follow Chen et al. (2021) that uses LMs for program synthesis to sample multiple output for each task. However, unlike prior works in program synthesis that choose the sample with highest unit test pass rate, we only consider the setting where one sample is allowed to be evaluated for each task. This is because repetitive trial-and-errors can be dangerous in the real world, and executing many action plans is equivalent to probing the environment for privileged information, which is often considered not viable.

### 3.2 ROBUST PARSING BY SEMANTIC TRANSLATION

One issue arises when naively following the above approach to generate action plans for high-level tasks: the action plan is often not executable because LMs are allowed to generate free-form text.

Therefore, most of the time the output cannot be mapped to one unambiguous actionable step. And many reasons can cause such failures: 1) the output does not follow pre-defined mappings of any atomic action (i.e. "I first walk to the bedroom" does not follow "Walk to ⟨PLACE⟩"), 2) the output may refer to atomic action and objects using words unrecognizable by the environment (i.e. "Clean the dirty dishes in the sink" where "clean" and "dirty dishes in the sink" cannot be mapped to precise action and object), 3) the output contains lexical ambiguous words (i.e. "Open TV" should instead be "Switch on TV"), or 4) the output may use disallowed action (i.e. "Microwave the cup").

Instead of developing a set of rules to transform the free-form text into admissible action steps, we propose to again leverage world knowledge learned by large language models to semantically translate the action. For each step in the action plan $\hat{a}$, we aim to find the most similar admissible environment action $a_e$ as measured by cosine similarity:

$$\underset{a_e}{\operatorname{argmax}} \frac{f(\hat{a}) \cdot f(a_e)}{\|f(\hat{a})\| \|f(a_e)\|} \text{ where } f \text{ is an embedding function.}$$

To embed the output text and environment actions, we use a BERT-style LM (Devlin et al., 2018; Liu et al., 2019) trained with Sentence-BERT (Reimers & Gurevych, 2019) objective because of its suitability for sentence modeling. The sentence embedding is obtained by mean-pooling the last layer hidden states across all tokens. We refer to this LM as **Translation LM**. Note that this is a different LM than the GPT-style Planning LM discussed in the text so far. Using a single LM for both purposes could as well be possible and likely more efficient, but we leave such investigation to future works. While the set of actions in our environment is discrete and possible to exhaustively enumerate, sampling or projection can be employed in larger discrete or continuous action spaces.

Since Translation LM can guarantee the parsed action is allowed by the environment, we can trade-off semantic soundness of an LM step by how likely it can be mapped to an admissible action in the environment. This can be achieved by a simple modification to the scheme that we use to choose the *best* sample from the LM output. Instead of only using mean token log probability as a ranking metric, we choose the sample with the highest score calculated as $s = C + \beta \cdot logprob$, where $C$ is the cosine similarity to the closest allowed action and $\beta$ is a weighting parameter.

### 3.3 AUTOREGRESSIVE TRAJECTORY CORRECTION

Translating each step of the program after the entire program has been synthesized is analogous to open-loop planning and is subject to compounding errors. In practice, LLMs might output compounded instructions for a single step, even though it cannot be completed using one admissible action in the environment. To this end, we can instead interleave *plan generation* and *action translation* to allow for automatic trajectory correction. At each step, we first query Planning LM to generate $k$ samples for a single action. Then we calculate score $s$ for each sample using Translation LM and append the translated action to the unfinished text completion. This way all subsequent steps will be conditioned on admissible actions instead of free-form text output generated by Planning LM. Furthermore, we can use Translation LM to detect out-of-distribution actions, those outside the capabilities of a robot, and terminate a program early instead of mapping to a faulty action. This can be easily implemented by setting a threshold $\epsilon$ such that if $C_{\max}^t < \epsilon$ at step $t$, the program is terminated early. We empirically show this leads to better executability while maintaining similar correctness of the generated action plans.

### 3.4 DYNAMIC EXAMPLE SELECTION FOR IMPROVED KNOWLEDGE EXTRACTION

So far in the text, we always give the same example in the prompt for all evaluated high-level tasks. However, consider the task of "ordering pizza". Prompting LLMs with this task may give the assumption that the agent is initialized in front of a computer, and the LLMs may guide the agent to search for a pizza store and click "checkout my cart". Although these are reasonable and feasible in the real world, such assumption cannot always be made as these interactions may not be supported in simulated environments like VirtualHome. In fact, the closest series of actions that human experts give may be "walking to a computer", "switching on the computer", and "typing the keyboard". Without being finetuned on these data, LLMs would often fail at these tasks. To provide weak supervision at inference time, we propose to use Translation LM to select the most similar task from the *demonstration set* to be used as the example in the prompt. Specifically, we choose the task

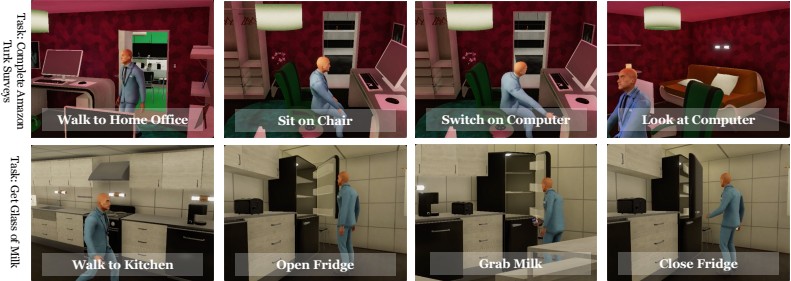

Figure 3: Visualization of VirtualHome programs generated by our approach. The top row shows the execution of the task "Complete Amazon Turk Surveys", and the bottom row shows the task "Get Glass of Milk". We show LLMs not only can generate sensible action plans given only high-level tasks but also contains the actionable knowledge that can be extracted for grounding in embodied environments.

whose high-level description matches closely the query task as measured by cosine similarity. This allows Planning LM to reason how to perform a similar task given a human-annotated example. In our evaluations, we make sure the demonstration set is not overlapping with our test queries.

Combining the various improvement discussed above, we refer to the final approach as **Translated [LM]**, where **[LM]** is replaced by specific language model used such as GPT-3 and Codex.

## 4 RESULTS

In this section, we first show that language models can generate sensible action plans for many high-level tasks, even without any additional training. Then we highlight its inadequacy when naively applied to embodied environments and demonstrate how this can be improved by again leveraging world knowledge learned by LLMs. Visualization of generated programs are shown in Fig 3.

**Sampling from LMs**  Pre-trained LMs are sensitive to sampling parameters and the specific example given in the prompt. For all evaluated methods, we perform hyper-parameter search over various sampling parameters, and for methods using fixed prompt example, we report metrics averaged across three randomly chosen examples. To select the best run for each method, we rank the runs by $LCS + executability$, each normalized by human-expert scores[3]. Further details can be found in Appendix 8.3.

**Model Choices**  We find empirically the combination of Codex-12B and Sentence-RoBERTa-355M work well in our setting. Codex and GPT-3 are accessed using OpenAI API. The remaining models are accessed through open-source packages, Hugging Face Transformers (Wolf et al., 2019) and SentenceTransformers (Reimers & Gurevych, 2019), without additional modifications.

### 4.1 DO LLMs CONTAIN ACTIONABLE KNOWLEDGE FOR HIGH-LEVEL TASKS?

We first investigate whether LLMs can generate sensible action plans expressed in free-form language. We use the approach described in Section 3.1 to query pre-trained LLMs. To evaluate the correctness of generated action plans, we conduct human evaluations. For each model, we ask 10 human annotators to determine – by answering "Yes" or "No" – whether each task can be completed using provided action steps. To provide a reference of how humans might rate the action plans provided by other humans, we also ask annotators to rate the ground-truth action plans provided in the VirtualHome dataset for the same set of tasks. In contrast to the free-form text output by LLMs, the ground-truth action plans from VirtualHome are generated via a graphical programming interface that enforces strict syntax, although annotators were allowed to compose necessary actions.

We show the human evaluation results in Fig 1, where y-axis shows correctness averaged across all tasks and all annotators. Surprisingly, when LLMs are large enough and without imposed syntactic constraints, they can generate highly realistic action plans whose correctness – as deemed by human annotators – even surpasses human-labeled ground-truth. Yet another interesting finding is

---

[3]See footnote 2.

| Language Model | Executability | LCS | Correctness |
|---|---|---|---|
| Vanilla GPT-2 117M | 18.66% | 3.19% | 14.27% |
| Vanilla GPT-2 1.5B | 39.40% | 7.78% | 19.51% |
| Vanilla Codex 2.5B | 17.62% | 15.57% | 53.44% |
| Vanilla GPT-Neo 2.7B | 29.92% | 11.52% | 50.74% |
| Vanilla Codex 12B | 18.07% | 16.97% | 56.06% |
| Vanilla GPT-3 12B | 25.87% | 13.40% | 30.98% |
| Vanilla GPT-3 175B | 7.79% | 17.82% | 65.19% |
| Annotated GT | 100.00% | N/A | 54.80% |
| Fine-tuned GPT-3 12B | 66.07% | 34.08% | 50.56% |
| *Our Final Methods* | | | |
| Translated Codex 12B | 78.57% | 24.72% | 34.97% |
| Translated GPT-3 175B | 73.05% | 24.09% | 51.73% |

Table 1: Human-evaluated correctness and evaluation results in VirtualHome. Although action plans generated by GPT-3 and Codex can even surpass the annotated GT in correctness measure, they are rarely executable. By translating the naive action plans, we show an important step towards grounding LLMs in embodied environments, but we observe room to achieve this without trading executability for correctness. We also observe a failure mode among smaller models that lead to high executability.

that Codex outperforms GPT-3 significantly under the same number of model parameters. One hypothesis could be that by fine-tuning on structured data (docstrings and code), Codex specializes at decomposing a high-level objective into a number of basic operations, even with those operations described in natural language. We also observe some level of correctness for smaller models such as GPT-2. However, inspection of its produced output indicates that it often generates significantly shorter plans by ignoring common-sense actions or by simply rephrasing the given task (e.g. the task "Go to sleep" produces only a single step "Go to bed"). These failure modes sometimes mislead human annotators to mark them correct as the annotators may ignore common-sense actions in their judgment as well, resulting in a higher correctness rate than the quality of the output shows.

## 4.2 How executable are the LLM action plans?

We analyze the executability of LLM plans by evaluating them in all 7 household scenes in VirtualHome. As shown in Table 1, we find action plans generated naively by LLMs are generally not very executable. Although smaller models seem to have higher executability, we find that the majority of these executable plans are produced by ignoring the queried task and repeating the given GT example of a different task. This is validated by the fact that smaller models have lower LCS than larger models despite having higher executability, showing that this failure mode is prevalent among smaller models. In contrast, larger models do not suffer severely from this failure mode. Yet as a result of being more expressive, their generated programs are substantially less executable.

## 4.3 Can LLM action plans be made executable by action translation?

In this section, we evaluate the effectiveness of our proposed procedure of action translation. We first create a bank of all allowed 47522 action steps in the environment, including all possible combinations of atomic actions and allowed arguments/objects. Then we use an off-the-shelf Sentence-RoBERTa (Liu et al., 2019; Reimers & Gurevych, 2019) as Translation LM to create embeddings for actions and output text. For better computational efficiency, we pre-compute the embeddings for all allowed actions, leaving minor computation overhead for our procedure over the baseline methods at inference time. As shown in Table 1, executability of generated programs is significantly improved. Furthermore, we also observe improved LCS because the translated action steps precisely follow the program syntax and thus are more similar to ground-truth annotations. One sample output is shown in Fig 1 and a larger random subset of generated samples can be found in Appendix 8.7.

To validate their correctness, we again perform human studies using the same procedure in Sec 4.1. Results are shown in Table 1. We find that despite being more similar to GT, the programs are deemed less correct by humans. By examining the generated output, we observe two main sources of errors. First, we find Translation LM is poor at mapping compounded instructions to a succinct

admissible action. This is partly due to that Translation LM is trained on a much smaller dataset and contains much a smaller number of parameters, so we expect further improvement by using a larger pre-trained model for translation. The second source of error comes from imperfect expressivity of the environment; we find that for many tasks we evaluate, certain necessary actions or objects are not implemented. This is also reflected by out human evaluation results of the GT programs, as only half of the programs are considered complete by the human annotators.

## 5 ANALYSIS AND DISCUSSIONS

### 5.1 ABLATION OF DESIGN DECISIONS

We perform ablation studies to show the effectiveness and necessity of three components of our proposed procedure, each described Sec 3.2, 3.3, and 3.4. As shown in Table 2, leaving out any of the three components would all

| Methods | Executability | LCS |
|---------|--------------|------|
| Translated Codex 12B | **78.57%** | **24.72%** |
| - **w/o** Action Translation | 31.49% | 22.53% |
| - **w/o** Dynamic Example | 50.86% | 22.84% |
| - **w/o** Iterative | 55.19% | 24.43% |

Table 2: Ablation of three proposed techniques.

lead to decreased performance in both executability and LCS. Notably, not doing action translation leads to the most significant executability drop, showing the importance of action translation in extracting executable action plans from LLMs.

### 5.2 CAN LLMS GENERATE ACTIONABLE PROGRAMS BY FOLLOWING DETAILED INSTRUCTIONS?

Prior works often focus on translating step-by-step instructions into executable programs. We evaluate LLMs under this setting using a prompt format shown in Appendix 8.2. Although this setting is easier as it does not require rich actionable knowledge, detailed instructions can help resolve much ambiguity of exactly how to perform a high-level task when multiple solutions are possible. Therefore, **Translated Codex 12B** achieves executability of $78.57\%$ and LCS of $32.87\%$, where LCS sees a considerable bump from the setting without detailed instructions. Surprisingly, the LCS result is very close to that of a supervised LSTM (Hochreiter & Schmidhuber, 1997) baseline from VirtualHome trained on human-annotated data, which is at $34.00\%$. Note that since code to train the baseline and the specific train/test split is not publicly released, we only show results reported in Puig et al. (2018) as a reference. We also cannot compare executability as it is not reported.

### 5.3 IS ACTION TRANSLATION NECESSARY FOR ACTIONABLE KNOWLEDGE GROUNDING?

The investigations of this paper are two-fold: 1) Is actionable knowledge present in LLMs? 2) Can we ground this actionable knowledge in interactive environment? In this section, we focus our attention on the second question by conditioning on the assumption that first question is true. To do this, since successful execution of *correct* action plans directly measures grounding, we select only the *correct* plans generated by LLMs and measure how executable they are. We deem an action plan to be *correct* if 70% or more human annotators decide it is correct.

| Methods | # of C | # of C and E | E / C |
|---------|--------|--------------|-------|
| GPT-2 1.5B | 8 | 0 | 0.00% |
| GPT-3 12B | 24 | 2 | 8.33% |
| GPT-3 175B | **68** | 5 | 7.35% |
| Codex 12B | 45 | 8 | 17.78% |
| Translated Codex 12B | 18 | **15** | **83.33%** |

Table 3: Count of correct/executable programs and percentage of executable among correct. C indicates correct and E indicates executable.

As shown by Table 3, when an LM is not large enough (e.g. GPT-2), not only it contains little actionable knowledge, but this knowledge cannot be grounded at all. GPT-3 and Codex, on the other hand, can generate highly correct action plans in free-form language. However, they do not have the capability to ground their actionable knowledge in interactive environments. What's more interesting, by comparing GPT-3 of both 12B parameters and 175B parameters, ratio of executable plans does not improve with the parameter count. This shows that simply training larger models does not necessarily lead to better knowledge grounding. In the meantime, action translation offers a promising way towards grounding actionable knowledge by producing highly executable plans. However, we again note that it comes at a trade-off of producing less correct plans as compared to its vanilla counterpart, and we hope to see future endeavors for bridging the gap.

# 6 RELATED WORKS

Large-scale natural language modeling has witnessed rapid advances since the inception of the Transformer architecture (Vaswani et al., 2017). It has been shown by recent works that large language models (LLMs) pre-trained on large unstructured text corpus not only can perform strongly on various down-stream NLP tasks (Devlin et al., 2018; Radford et al., 2019; Raffel et al., 2019; Brown et al., 2020) but also can internalize an implicit knowledge base containing rich information about the world (Petroni et al., 2019; Jiang et al., 2020; Davison et al., 2019; Talmor et al., 2020; Roberts et al., 2020). Furthermore, the learned representations can be used to model relations of entities (Li et al., 2021), retrieve matching visual features (Ilharco et al., 2020), and even as valuable priors when applied to diverse tasks from different modalities (Lu et al., 2021; Tsimpoukelli et al., 2021). Compared to prior works in knowledge extraction that extract *single-step factual answers* memorized by the models (e.g. "Dante was born in [PLACE]"), we aim to extract *sequential action plans* to complete an open-ended human activity (e.g. "make breakfast"). We further require these plans to only contain allowed actions and satisfy the pre/post-conditions of actions in order to be executed by an embodied agent.

At the same time, there has also been growing interest and development in grounding language in embodied environment. A series of prior works have investigated the possibility of parsing language instructions into formal logic to resolve various linguistic ambiguities for embodied agents (Artzi & Zettlemoyer, 2013; Misra et al., 2015; Tenorth et al., 2010). However, they often scale poorly to complex tasks and environments. Recently, more research efforts have been put into creating better and more realistic environments with the goal to further advances in this area (Puig et al., 2018; Shridhar et al., 2020a;b; Kolve et al., 2017; Savva et al., 2019). At the same time, by leveraging the better representation power of neural architectures, a number of works have looked into creating instruction-following agents that can perform manipulation (Lynch & Sermanet, 2020), navigation (Majumdar et al., 2020), or both (Suglia et al., 2021; Hill et al., 2020).

Notably, most of these prior works do not leverage full-blown pre-trained LLMs (Suglia et al., 2021) or do not scale to complex human activities (Hill et al., 2020; Lynch & Sermanet, 2020). Perhaps more importantly, few works have evaluated LLMs in an embodiment setting that realizes the full potential of the world knowledge these models contain: the tasks evaluated are often "pick", "grab", "open", and etc, which do not resemble the highly diverse activities that humans perform in daily lives. The development of VirtualHome environment Puig et al. (2018) enables such possibility. However, relevant works (Puig et al., 2020; Liao et al., 2019) rely on human-annotated data and perform supervised training from scratch. Due to the lack of rich world knowledge, these models can only generate action plans given step-by-step instructions of how to act or video demonstrations. In this work, we take a step further by conditioning only on the high-level descriptions and by extracting executable action plans from LLMs without any additional training.

# 7 LIMITATIONS AND CONCLUSION

There are several notable limitations of this work. First, although our approach presents a viable way to ground world knowledge in embodied environments, it is still a trade-off rather than one best solution since we observe considerable drop in correctness. Second, we focus on high-level to mid-level grounding, assuming there is a controller that can execute mid-level tasks (such as "grab cup"). Our work does not investigate usefulness of LLMs for low-level sensorimotor behavior grounding. The third limitation is that we do not incorporate observation context or feedback into our models. To some extent, we approach LLMs in the same way as how VirtualHome asks human annotators to give action plans for a given huamn activity by *imagination*, in which case the human-generated action plans also do not incorporate observation context. However, we do see incorporating observation context for complex activities as an exciting future direction.

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

## 8 APPENDIX

### 8.1 EXAMPLE PROGRAM IN VIRTUALHOME

[Walk] ⟨ living_room ⟩ (1)
[Walk] ⟨ television ⟩ (1)
[Find] ⟨ television ⟩ (1)
[SwitchOn] ⟨ television ⟩ (1)
[Find] ⟨ sofa ⟩ (1)
[Sit] ⟨ sofa ⟩ (1)
[TurnTo] ⟨ television ⟩ (1)
[Watch] ⟨ television ⟩ (1)

Table 4: An example program for the activity "Relax on sofa".

### 8.2 EXAMPLE PROMPT CONTAINING STEP-BY-STEP INSTRUCTIONS

| | |
|---|---|
| Task: Read book | Step 8: Sit on chair |
| Description: Walk to home office, turn on light, grab a book, sit in chair, start to read the book. | Step 9: Read novel |
| Step 1: Walk to home office | Task: Find dictionary |
| Step 2: Walk to light | Description: Move towards the |
| Step 3: Find light | bookshelf, scan the bookshelf for |
| Step 4: Switch on light | the dictionary, when the |
| Step 5: Find novel | dictionary is found, pick up the |
| Step 6: Grab novel | dictionary. |
| Step 7: Find chair | |

Figure 4: An example prompt containing step-by-step instructions.

### 8.3 HYPERPARAMETER SEARCH

For each evaluated method, we perform grid search over the following hyperparameters:

| Name | Description | Search Values |
|---|---|---|
| epsilon ($\epsilon$) | OOD cutoff threshold used in iterative action translation | $\{0, 0.4, 0.8\}$ |
| temperature | sampling parameter adjusting relative probabilities across tokens | $\{0.1, 0.3, 0.6\}$ |
| k | number of samples generated when querying LMs each time | $\{1, 10\}$ |
| frequence_penalty | OpenAI API specific; penalize new tokens based on their existing frequency in the text so far | $\{0.1, 0.3, 0.6, 0.9\}$ |
| presence_penalty | OpenAI API specific; penalize new tokens based on whether they appear in the text so far | $\{0.3, 0.5, 0.8\}$ |
| repetition_penalty | Hugging Face Transformers specific; penalize new tokens based on whether they are repeating existing text | $\{1.0, 1.2, 1.5, 1.8\}$ |

For methods that use fixed example across evaluated tasks, we search over the following three randomly chosen examples:

| Example 1 | Example 2 | Example 3 |
|---|---|---|
| Task: Use computer
Step 1: Walk to home office
Step 2: Walk to chair
Step 3: Find chair
Step 4: Sit on chair
Step 5: Find computer
Step 6: Switch on computer
Step 7: Turn to computer
Step 8: Look at computer
Step 9: Find keyboard
Step 10: Type on keyboard | Task: Relax on sofa
Step 1: Walk to home office
Step 2: Walk to couch
Step 3: Find couch
Step 4: Sit on couch
Step 5: Find pillow
Step 6: Lie on couch | Task: Read book
Step 1: Walk to home office
Step 2: Walk to novel
Step 3: Find novel
Step 4: Grab novel
Step 5: Find chair
Step 6: Sit on chair
Step 7: Read novel |

## 8.4 ANALYSIS OF PROGRAM LENGTH

Shorter programs have a natural advantage of being more executable. Consider a task "wash hands" and a corresponding program that only commands an agent to "go to bathroom" without additional steps. The program is obviously incorrect yet trivially executable. To validate our approach does not simply generate very short program, we calculate the average program length across the 88 evaluated tasks. Results are shown in Table 6. In addition to the failure mode discussed in Section 4.2 that leads to incorrect yet executable programs, smaller LMs such as GPT-2 also generate programs significantly shorter than larger models, making them more executable. In contrast, larger models like Codex-12B generate more expressive program of high correctness, but they often suffer from executability. We show action translation can lead to benefits of both worlds, generating programs that are highly executable while maintaining similar expressiveness in terms of program length.

| Methods | Executability | Average Length |
|---|---|---|
| Vanilla GPT-2 1.5B | 39.40% | 4.24 |
| Vanilla Codex 12B | 18.07% | 7.22 |
| Translated Codex 12B | 78.57% | 7.13 |
| Ground-Truth | 100.00% | 9.66 |

Table 6: Average executability & program length of different methods and human annotations.

## 8.5 DETAILS OF HUMAN EVALUATIONS

Human evaluations are conducted on Amazon Mechanical Turk. For each method, we generate action plans for all 88 high-level tasks. To account for expressivity of the VirtualHome environment (Puig et al., 2018), we further include ground-truth action plans from the VirtualHome dataset in our human evaluations. The human evaluations are conducted in the form of questionnaires containing all action plans with unknown corresponding methods. The questionnaire contains the following instructions at the top:

> *For every question below, determine whether the task can be completed in any reasonable scenario using the provided steps. In other words, can the task be decomposed into these steps? Note that simply re-stating the task does not mean completing it.*

Human annotators are required to answer all the questions in the questionnaire, where each question is an action plan generated by a method unknown to the annotator. The order of the questions is randomly permuted before presented to each annotator. For each question, the annotators need to answer either "Yes" or "No" indicating if they believe the action plan completes the task. For each method, we report *correctness* percentage averaged across 10 participated human annotators.

## 8.6 DETAILS OF VIRTUALHOME TASKS

**VirtualHome** *ActivityPrograms*   is a knowledge base collected on Mechanical Turk by Puig et al. (2018). The knowledge base features real household activities of humans and correspondingly sequences of **basic actions** for robots to perform these activities. Each entry contains a high-level task name (i.e. "Watch TV"), a mid-level detailed description (i.e. "Sit on my couch directly opposite my TV, switch on my TV with the remote control and watch"), and a sequence of allowed actions (a.k.a. a program) to perform the task in VirtualHome. There are 292 distinct high-level tasks and a total of 2821 different entries in the knowledge base, where each high-level task may correspond one or several programs. The knowledge base covers a diverse set of real human activities and a wide range of different actions and objects. Puig et al. (2018) also conduct additional human evaluation on 100 randomly sampled programs and show that 64% of them are complete (i.e. contain all necessary steps). Readers are encouraged to refer to Puig et al. (2018) for more comprehensive analysis of the dataset.

**Train/Test Split**   From all 292 distinct high-level tasks from the *ActivityPrograms* knowledge base, we randomly sample 88 high-level tasks for evaluation. The remaining 204 tasks are used as *demonstration set* from which we are allowed to select as example(s) for prompting language models, or in the case of supervised fine-tuning, they are used to fine-tune pre-trained language models.

## 8.7 RANDOM SAMPLES OF GT, VANILLA CODEX, AND TRANSLATED CODEX

| Annotated Ground-Truth | Codex | Translated Codex (Ours) |
|---|---|---|
| Task: Go to sleep
Step 1: Walk to bedroom
Step 2: Walk to bed
Step 3: Find bed
Step 4: Turn to bed
Step 5: Look at bed
Step 6: Sit on bed
Step 7: Sleep | Task: Go to sleep
Step 1: Walk to bedroom
Step 2: Walk to bed
Step 3: Find bed
Step 4: Sit on bed
Step 5: Find pillow
Step 6: Sleep on pillow | Task: Go to sleep
Step 1: Walk to bedroom
Step 2: Walk to bed
Step 3: Lie on bed
Step 4: Sleep |

| | | |
|---|---|---|
| Task: Browse internet
Step 1: Walk to home office
Step 2: Walk to chair
Step 3: Find chair
Step 4: Sit on chair
Step 5: Find computer
Step 6: Switch on computer
Step 7: Turn to computer
Step 8: Look at computer
Step 9: Find keyboard
Step 10: Type on keyboard | Task: Browse internet
Step 1: Walk to home office
Step 2: Walk to chair
Step 3: Find chair
Step 4: Sit on chair
Step 5: Find computer
Step 6: Switch on computer
Step 7: Turn to computer
Step 8: Look at computer
Step 9: Find keyboard
Step 10: Type on keyboard | Task: Browse internet
Step 1: Walk to home office
Step 2: Walk to desk
Step 3: Find chair
Step 4: Sit on chair
Step 5: Find computer
Step 6: Switch on computer
Step 7: Find mouse
Step 8: Grab mouse
Step 9: Turn to computer
Step 10: Point at computer |
| Task: Listen to music
Step 1: Walk to bedroom
Step 2: Walk to computer
Step 3: Find computer
Step 4: Turn to computer
Step 5: Look at computer
Step 6: Walk to computer
Step 7: Find chair
Step 8: Pull chair
Step 9: Sit on chair
Step 10: Find headset
Step 11: Grab headset
Step 12: Put on headset
Step 13: Switch on computer
Step 14: Find keyboard
Step 15: Type on keyboard | Task: Listen to music
Step 1: Walk to home office
Step 2: Walk to chair
Step 3: Find chair
Step 4: Sit on chair
Step 5: Find computer
Step 6: Switch on computer
Step 7: Turn to computer
Step 8: Look at computer
Step 9: Find keyboard
Step 10: Type on keyboard | Task: Listen to music
Step 1: Walk to home office
Step 2: Walk to computer
Step 3: Sit on chair
Step 4: Switch on computer
Step 5: Find music stand |
| Task: Brush teeth
Step 1: Walk to bathroom
Step 2: Walk to tooth paste
Step 3: Find tooth paste
Step 4: Grab tooth paste
Step 5: Find toothbrush
Step 6: Grab toothbrush | Task: Brush teeth
Step 1: Walk to bathroom
Step 2: Walk to sink
Step 3: Find toothbrush
Step 4: Pick up toothbrush
Step 5: Put toothbrush in mouth
Step 6: Move brush around mouth for two minutes
Step 7: Spit out toothpaste and brush into sink
Step 8: Turn on water in sink and rinse brush for one minute
Step 9: Turn off water in sink and return brush to cupboard | Task: Brush teeth
Step 1: Walk to bathroom
Step 2: Open door
Step 3: Walk to sink
Step 4: Put pot on sink
Step 5: Put brush on toothbrush
Step 6: Turn to toothpaste
Step 7: Put toothpaste on toothbrush
Step 8: Put teeth on toothbrush |
| Task: Dry soap bottles
Step 1: Find towel
Step 2: Find soap
Step 3: Grab soap
Step 4: Wipe soap
Step 5: Put back soap
Step 6: Find shampoo
Step 7: Grab shampoo
Step 8: Wipe shampoo
Step 9: Put back shampoo
Step 10: Find shampoo
Step 11: Grab shampoo
Step 12: Wipe shampoo
Step 13: Put back shampoo | Task: Dry soap bottles
Step 1: Walk to kitchen
Step 2: Walk to sink
Step 3: Find sink
Step 4: Turn on water tap
Step 5: Wait for water to be hot enough
Step 6: Fill container with hot water
Step 7: Find soap bottles
Step 8: Pour soap into container with hot water
Step 9: Wait for soap to dissolve in water
Step 10: Turn off tap and empty container into sink | Task: Dry soap bottles
Step 1: Walk to bathroom
Step 2: Walk to shower
Step 3: Find soap |

| | | |
|---|---|---|
| Task: Vacuum
Step 1: Walk to home office
Step 2: Walk to computer
Step 3: Find vacuum cleaner
Step 4: Switch on vacuum cleaner
Step 5: Switch off vacuum cleaner | Task: Vacuum
Step 1: Walk to hallway
Step 2: Walk to closet
Step 3: Find closet
Step 4: Open closet
Step 5: Find vacuum cleaner
Step 6: Turn on vacuum cleaner
Step 7: Walk to living room
Step 8: Vacuum floor in living room | Task: Vacuum
Step 1: Walk to home office
Step 2: Walk to dresser
Step 3: Open dresser
Step 4: Find vacuum cleaner
Step 5: Grab vacuum cleaner
Step 6: Pull vacuum cleaner
Step 7: Plug in vacuum cleaner
Step 8: Switch on vacuum cleaner
Step 9: Pull vacuum cleaner
Step 10: Push vacuum cleaner |
| Task: Find dictionary
Step 1: Walk to home office
Step 2: Walk to bookshelf
Step 3: Turn to bookshelf
Step 4: Look at bookshelf
Step 5: Turn to bookshelf
Step 6: Look at bookshelf
Step 7: Turn to bookshelf
Step 8: Look at bookshelf
Step 9: Turn to bookshelf
Step 10: Look at bookshelf
Step 11: Turn to bookshelf
Step 12: Look at bookshelf
Step 13: Turn to bookshelf
Step 14: Look at bookshelf
Step 15: Find novel
Step 16: Grab novel | Task: Find dictionary
Step 1: Walk to home office
Step 2: Walk to chair
Step 3: Find chair
Step 4: Sit on chair
Step 5: Find dictionary
Step 6: Open dictionary
Step 7: Look at dictionary | Task: Find dictionary
Step 1: Walk to home office
Step 2: Walk to book
Step 3: Find textbook |
| Task: Wash monitor
Step 1: Walk to home office
Step 2: Walk to computer
Step 3: Find computer
Step 4: Turn to computer
Step 5: Look at computer
Step 6: Switch off computer
Step 7: Find brush
Step 8: Turn to brush
Step 9: Look at brush
Step 10: Grab brush
Step 11: Wipe computer
Step 12: Drop brush
Step 13: Find napkin
Step 14: Grab napkin
Step 15: Wash napkin
Step 16: Wipe computer | Task: Wash monitor
Step 1: Walk to home office
Step 2: Walk to chair
Step 3: Find chair
Step 4: Sit on chair
Step 5: Find computer
Step 6: Switch on computer
Step 7: Turn to computer
Step 8: Look at computer
Step 9: Find keyboard
Step 10: Type on keyboard | Task: Wash monitor
Step 1: Walk to check
Step 2: Walk to desk |
| Task: Read newspaper
Step 1: Walk to home office
Step 2: Walk to chair
Step 3: Find chair
Step 4: Sit on chair
Step 5: Find newspaper
Step 6: Grab newspaper
Step 7: Open newspaper
Step 8: Read newspaper | Task: Read newspaper
Step 1: Walk to home office
Step 2: Walk to chair
Step 3: Find chair
Step 4: Sit on chair
Step 5: Find newspaper
Step 6: Open newspaper
Step 7: Read newspaper | Task: Read newspaper
Step 1: Walk to home office
Step 2: Walk to chair
Step 3: Find chair
Step 4: Sit on chair
Step 5: Find newspaper
Step 6: Grab newspaper
Step 7: Read newspaper |

| | | |
|---|---|---|
| Task: Apply lotion
Step 1: Find bed
Step 2: Turn to bed
Step 3: Point at bed
Step 4: Wash bed
Step 5: Turn to bed
Step 6: Look at bed
Step 7: Push bed
Step 8: Sit on bed | Task: Apply lotion
Step 1: Walk to bathroom
Step 2: Walk to cabinet
Step 3: Open cabinet
Step 4: Find lotion
Step 5: Apply lotion on hands
Step 6: Rub hands together for 10 seconds
Step 7: Apply lotion on face | Task: Apply lotion
Step 1: Walk to bathroom
Step 2: Turn to lotionbottle
Step 3: Grab lotionbottle |
| Task: Draw picture
Step 1: Walk to home office
Step 2: Walk to chair
Step 3: Find chair
Step 4: Sit on chair
Step 5: Find desk
Step 6: Open desk
Step 7: Find check
Step 8: Grab check
Step 9: Put check on desk
Step 10: Find pencil
Step 11: Grab pencil
Step 12: Close desk | Task: Draw picture
Step 1: Walk to home office
Step 2: Walk to chair
Step 3: Find chair
Step 4: Sit on chair
Step 5: Find computer
Step 6: Switch on computer
Step 7: Turn to computer
Step 8: Look at computer
Step 9: Find keyboard
Step 10: Type on keyboard | Task: Draw picture
Step 1: Walk to home office
Step 2: Walk to wall
Step 3: Walk to drawing
Step 4: Find drawing
Step 5: Turn to drawing
Step 6: Look at drawing
Step 7: Touch drawing
Step 8: Pull drawing
Step 9: Push drawing |
| Task: Prepare pot of boiling water
Step 1: Walk to dining room
Step 2: Walk to pot
Step 3: Find pot
Step 4: Grab pot
Step 5: Walk to sink
Step 6: Find faucet
Step 7: Switch on faucet
Step 8: Put pot on sink
Step 9: Grab pot
Step 10: Switch off faucet
Step 11: Grab pot
Step 12: Walk to oven
Step 13: Put pot on oven
Step 14: Switch on oven | Task: Prepare pot of boiling water
Step 1: Walk to kitchen
Step 2: Walk to sink
Step 3: Find sink
Step 4: Turn on tap
Step 5: Fill kettle with water
Step 6: Put kettle on stove
Step 7: Turn on stove
Step 8: Wait for water to boil | Task: Prepare pot of boiling water
Step 1: Walk to kitchen
Step 2: Walk to cabinet
Step 3: Find pot
Step 4: Grab pot
Step 5: Walk to stove
Step 6: Find stove
Step 7: Switch on stove |
| Task: Look at mirror
Step 1: Walk to bathroom
Step 2: Walk to mirror
Step 3: Find mirror
Step 4: Turn to mirror
Step 5: Look at mirror | Task: Look at mirror
Step 1: Walk to bathroom
Step 2: Walk to mirror
Step 3: Look at mirror | Task: Look at mirror
Step 1: Walk to bedroom
Step 2: Walk to mirror
Step 3: Find mirror
Step 4: Turn to mirror
Step 5: Look at mirror |
| Task: Wipe down sink
Step 1: Walk to dining room
Step 2: Walk to rag
Step 3: Find rag
Step 4: Grab rag
Step 5: Walk to sink
Step 6: Wipe sink
Step 7: Put back rag | Task: Wipe down sink
Step 1: Walk to kitchen
Step 2: Walk to sink
Step 3: Find sink
Step 4: Wipe down sink
Step 5: Return to kitchen | Task: Wipe down sink
Step 1: Walk to bathroom
Step 2: Walk to cleaning solution
Step 3: Find cleaning solution
Step 4: Grab cleaning solution
Step 5: Find sink
Step 6: Pour cleaning solution into glasses
Step 7: Find faucet
Step 8: Switch on faucet
Step 9: Scrub sink
Step 10: Wash sink |

| | | |
|---|---|---|
| Task: Complete surveys on amazon turk
Step 1: Walk to chair
Step 2: Sit on chair
Step 3: Find computer
Step 4: Switch on computer
Step 5: Turn to computer
Step 6: Look at computer
Step 7: Find mouse
Step 8: Grab mouse
Step 9: Turn to computer
Step 10: Point at computer
Step 11: Find check
Step 12: Grab check
Step 13: Read check
Step 14: Put back mouse
Step 15: Find keyboard
Step 16: Type on keyboard | Task: Complete surveys on amazon turk
Step 1: Walk to home office
Step 2: Walk to chair
Step 3: Find chair
Step 4: Sit on chair
Step 5: Find computer
Step 6: Switch on computer
Step 7: Turn to computer
Step 8: Look at computer
Step 9: Find keyboard
Step 10: Type on keyboard | Task: Complete surveys on amazon turk
Step 1: Walk to home office
Step 2: Walk to desk
Step 3: Find chair
Step 4: Sit on chair
Step 5: Find computer
Step 6: Switch on computer |
| Task: Write book
Step 1: Walk to home office
Step 2: Walk to laptop
Step 3: Find laptop
Step 4: Grab laptop
Step 5: Find electrical outlet
Step 6: Plug in laptop
Step 7: Walk to love seat
Step 8: Sit on love seat
Step 9: Switch on laptop
Step 10: Find keyboard
Step 11: Type on keyboard | Task: Write book
Step 1: Walk to home office
Step 2: Walk to chair
Step 3: Find chair
Step 4: Sit on chair
Step 5: Find computer
Step 6: Switch on computer
Step 7: Turn to computer
Step 8: Look at computer
Step 9: Find keyboard
Step 10: Type on keyboard | Task: Write book
Step 1: Walk to home office
Step 2: Walk to light
Step 3: Find light
Step 4: Switch on light
Step 5: Find pen
Step 6: Grab pen
Step 7: Find chair
Step 8: Sit on chair
Step 9: Turn to novel |
| Task: Paint ceiling
Step 1: Find drawing
Step 2: Turn to drawing
Step 3: Point at drawing
Step 4: Watch drawing
Step 5: Touch drawing
Step 6: Walk to bedroom | Task: Paint ceiling
Step 1: Walk to home office
Step 2: Walk to chair
Step 3: Find chair
Step 4: Sit on chair
Step 5: Find paint can
Step 6: Open paint can
Step 7: Pour paint into paint can
Step 8: Close paint can
Step 9: Find roller
Step 10: Roll paint onto ceiling | Task: Paint ceiling
Step 1: Walk to home office
Step 2: Walk to ceiling
Step 3: Walk to painting
Step 4: Find painting
Step 5: Turn to painting
Step 6: Look at painting
Step 7: Touch painting
Step 8: Pull painting
Step 9: Push painting |
| Task: Change clothes
Step 1: Walk to bedroom
Step 2: Walk to dresser
Step 3: Open dresser
Step 4: Find dress
Step 5: Grab dress
Step 6: Put on dress
Step 7: Close dresser | Task: Change clothes
Step 1: Walk to closet
Step 2: Open closet
Step 3: Find clothes
Step 4: Put on clothes
Step 5: Close closet | Task: Change clothes
Step 1: Walk to bedroom
Step 2: Walk to dresser
Step 3: Find dresser
Step 4: Open dresser
Step 5: Find pants
Step 6: Grab pants
Step 7: Put on pants
Step 8: Find shirt
Step 9: Grab shirt
Step 10: Put on shirt
Step 11: Close dresser |

| | | |
|---|---|---|
| Task: Make bed
Step 1: Walk to bedroom
Step 2: Walk to bed
Step 3: Find sheets
Step 4: Grab sheets
Step 5: Pull sheets | Task: Make bed
Step 1: Walk to bedroom
Step 2: Walk to bed
Step 3: Find bed
Step 4: Sit on bed
Step 5: Find sheet
Step 6: Pick up sheet
Step 7: Find pillow
Step 8: Pick up pillow
Step 9: Find blanket
Step 10: Pick up blanket | Task: Make bed
Step 1: Stand up
Step 2: Walk to bedroom
Step 3: Walk to bed
Step 4: Turn to bed
Step 5: Put sheets on bed |
| Task: Read yourself to sleep
Step 1: Walk to bedroom
Step 2: Walk to novel
Step 3: Find novel
Step 4: Grab novel
Step 5: Find bed
Step 6: Sit on bed
Step 7: Find bookmark
Step 8: Grab bookmark
Step 9: Read novel
Step 10: Put bookmark on novel
Step 11: Sleep | Task: Read yourself to sleep
Step 1: Walk to home office
Step 2: Walk to chair
Step 3: Find chair
Step 4: Sit on chair
Step 5: Find computer
Step 6: Switch on computer
Step 7: Turn to computer
Step 8: Look at computer
Step 9: Find keyboard
Step 10: Type on keyboard | Task: Read yourself to sleep
Step 1: Walk to bedroom
Step 2: Walk to book
Step 3: Sit on chair
Step 4: Read check
Step 5: Sleep |
| Task: Hang keys
Step 1: Walk to bedroom
Step 2: Walk to dresser
Step 3: Find purse
Step 4: Grab purse
Step 5: Open purse
Step 6: Find keys
Step 7: Grab keys
Step 8: Find bowl
Step 9: Put keys on bowl
Step 10: Close purse
Step 11: Put back purse | Task: Hang keys
Step 1: Walk to home office
Step 2: Walk to chair
Step 3: Find chair
Step 4: Sit on chair
Step 5: Find computer
Step 6: Switch on computer
Step 7: Turn to computer
Step 8: Look at computer
Step 9: Find keyboard
Step 10: Type on keyboard | Task: Hang keys
Step 1: Walk to home office
Step 2: Walk to table
Step 3: Find keys
Step 4: Grab keys
Step 5: Walk to wall
Step 6: Find hanger
Step 7: Put keys on hanger |
| Task: Eat cheese
Step 1: Walk to dining room
Step 2: Walk to freezer
Step 3: Open freezer
Step 4: Find cheese
Step 5: Grab cheese
Step 6: Eat cheese | Task: Eat cheese
Step 1: Walk to fridge
Step 2: Open fridge
Step 3: Find cheese
Step 4: Pick up cheese
Step 5: Walk to table
Step 6: Place cheese on table
Step 7: Close fridge
Step 8: Walk to chair
Step 9: Sit on chair
Step 10: Eat cheese | Task: Eat cheese
Step 1: Walk to dining room
Step 2: Walk to table
Step 3: Find table
Step 4: Turn to table
Step 5: Find chair
Step 6: Sit on chair
Step 7: Find food
Step 8: Grab food
Step 9: Find plate
Step 10: Put food on plate |

