# OpenReview forum: "Language Models as Zero-Shot Planners: Extracting Actionable Knowledge for Embodied Agents"
_ICLR.cc/2022/Conference — ICLR 2022 Submitted_

### Official Review · Reviewer_gxEx · 2021-10-31

**Correctness:** 4
**Technical Novelty And Significance:** 4
**Empirical Novelty And Significance:** 4
**Recommendation:** 6
**Confidence:** 4

**Main Review:**

Strengths:
* The paper studies an idea that seems relatively novel to this reviewer -- how well can models generate action plans in an embodied environment (not just code, or language)?
* The paper presents a simple idea -- using a translation LM to constrain the generated actions -- that greatly improves performance, by ensuring that the actions generated by the LM are valid within the environment.
* The paper has results that consider LMs of a variety of different sizes, from 1.5B to 175B, including both GPT-3 and codex families. The results seem to suggest that size is important, at least in this setting, insofar as the smaller models just copy from the input prompt.
* This paper presents analysis about the kinds of programs generated by LMs which I think might help future work build off of this direction.

Weaknesses:
* The experimental setup proposed by this paper for measuring grounding might not be ideal. In the VirtualHome setup, the action space is composed of mid-level actions (like "walk to home office") versus low-level actions ("walk forward 1m" ,etc), so the promise of LMs for embodied understanding might not generalize to more difficult environents (like Alfred; Shridhar et al 2020a). That said, I think this limitation is discussed well in the limitations section (Sec7).
* Though I appreciate the authors' work on making the evaluation robust, it is not clear to this reviewer that it is a good measure of correctness. To the best of this reviewer's understanding (of Section 2.2; Correctness), in the VirtualHome environment that the authors study, it is nontrivial to compare whether the final environment state of a generated program is similar to the "ground truth" environment state. The details of the human evaluation are a bit unclear, but it seems like humans are just given the list of actions rather than the world state after executing those actions. In this setup, it seems like there is a very strong bias towards language that 'looks like' real action plans, rather than language that is truly actionable. The result is that models outperform humans at the task. To this reviewer, the paper could be improved if the correctness measured something grounded, rather than just something that looks grounded.


Clarification:
* I'd appreciate more details on the human evaluation somewhere (like the interface being shown to the turkers)

**Summary Of The Paper:**

This paper studies left-to-right language models for on the VirtualHome planning task. In this task, a model is given a high level goal like "get glass of milk" and it must generate a sequence of lower-level steps to execute that goal ('walk to kitchen', 'open fridge', etc.). These steps correspond to the annotated scratch programs built by crowd workers as part of VirtualHome, so as such they are executable plans to achieve the given goal.

This paper uses a left-to-right LM to generate those lower-level steps, given one prompted example. The 1.5B-sized LMs do not seem to do very well at generating actions given this prompt. The larger 12B, etc. LMs do better, but frequently deviate from the set of valid actions, so the authors introduce a Translation LM that measures the similarity between any "action" that the LM generates and the set of valid actions. At each step, this translation LM is used in the loop to constrain the chosen action, and this improves executability significantly (e.g. 7% -> 73% for GPT-3 and 18% -> 78% for Codex). Helpful analysis breaks down this performance.

**Summary Of The Review:**

Overall, my opinions on this paper are fairly positive, because it tackles an interesting new direction with new simple methodology for generating actions from LMs. However, there are some issues with evaluation, that could greatly improve the paper if addressed (or at the least, discussed a bit more in the limitations section).

--

Update post review: I am lowering my score from 8 -> 6. I still think this is a good paper and it should be accepted, but I am not very convinced by the evaluation. I think the evaluation issues I (and KhzN) have issues with could be improved between acceptance and publication though.

---

> ### Author Response · Authors · 2021-11-21
> **Response to Reviewer gxEx**
>
> Thank you for the constructive feedback! Please find below for answers and clarifications to your comments, and please also refer to our response to common questions and concerns above.
>
> > *“The experimental setup proposed by this paper for measuring grounding might not be ideal… so the promise of LMs for embodied understanding might not generalize to more difficult environments (like Alfred; Shridhar et al 2020a)"*
> - Our goal is to study if the knowledge already contained in the language models can be extracted to achieve high-level tasks for embodied agents given access to low-level controllers, *without fine-tuning on environment-specific data*.
> - Therefore, instead of strongly grounding the LLM generation by using downstream data from the environment, we focused on grounding with weak constraints such that we evaluate LLMs raw knowledge as closely as possible.
> - To generate action plans for complex human activities such as `make the bed` and `make coffee`,  we’d like the weak constraints to ensure these plans obey common-sense knowledge, which is measured by the environment pre/post-conditions. By satisfying these pre/post-conditions, an action plan can indeed be executed in an embodied environment such as VirtualHome. For an atomic action `open`, one pre-condition is that the agent cannot grab milk from the fridge before opening it, and one post-condition is that the state of the fridge changes from “closed” to “open”.
> - While we believe the LLM actionable knowledge is also useful for environments like ALFRED, we do not investigate ALFRED because it requires object mask prediction. Due to the same reason, additional environment-specific data and a carefully designed architecture are likely needed, which conflates our contributions relating to raw knowledge in LLMs. Having said that, we look forward to future works that investigate how to utilize actionable knowledge in LLMs for these environments.
>
> > *“[In human evaluations,] it seems like humans are just given the list of actions rather than the world state after executing those actions… I'd appreciate more details on the human evaluation somewhere"*
>
> - Yes. In the conducted human evaluations, humans are only given the actions without any environment execution/visualization.
> - As suggested, we have updated the paper with additional details about human evaluations in the Appendix. A briefer version is provided below.
> - Human evaluations are conducted on Amazon Mechanical Turk. For each method, we generate action plans for all 88 high-level tasks. The evaluations are conducted in the form of questionnaires containing all action plans with *unknown corresponding methods* and with *randomly permuted order*. The questionnaire contains the following instructions at the top:
>
> ``
> For every question below, determine whether the task can be completed in any reasonable scenario using the provided steps. In other words, can the task be decomposed into these steps? Note that simply re-stating the task does not mean completing it.
> ``
>
> - Human annotators are required to provide “Yes” or “No” to all the questions in the questionnaire, indicating if they believe the action plan completes the task. For each method, we report the correctness percentage averaged across 10 participating human annotators.
>
> > *“the paper could be improved if the correctness measured something grounded, rather than just something that looks grounded."*
>
> - Unfortunately, due to the open-ended nature of the tasks we investigate in this paper, to the best of our knowledge, there isn’t a single straightforward metric that one can computationally produce to measure this. A metric that would serve a similar purpose would be “the percentage of action plans that are both correct and executable”, which is briefly discussed in Section 5.3 of the paper. However, we do not focus our investigations on this because it entangles what are effectively two different metrics. Same as Reviewer 28Tw, we believe that evaluating two axes separately can “better characterize the strengths and limitations of the language models studied in this paper” (28Tw), which can guide future efforts to determine what aspects to work on.
>
> #### References
>
> [1] ALFRED: A Benchmark for Interpreting Grounded Instructions for Everyday Tasks. Mohit Shridhar and Jesse Thomason and Daniel Gordon and Yonatan Bisk and Winson Han and Roozbeh Mottaghi and Luke Zettlemoyer and Dieter Fox. CVPR2020.

---

> > ### Comment · Reviewer_gxEx · 2021-11-25
> > **thanks for the response!**
> >
> > thanks for the response!
> >
> > The evaluation details in the paper revision are somewhat helpful, but I am still not sure I fully understand the interface. To this reviewer, the paper could be improved if you provided a screenshot of it.
> >
> > I still believe (agreeing with reviewer KhzN) that the evaluation could be improved. To this reviewer, the response isn't very convincing (e.g. sure -- a task like brewing coffee is open-ended, but there are certain necessary steps that you'd need to do to carry it out). Possibly, giving the workers some notion of a 'ground truth' trajectory (aka the target amount of detail that you're looking for) would help address this issue.

---

> > > ### Author Response · Authors · 2021-11-30
> > > **Response to Reviewer gxEx**
> > >
> > > Thanks for the reply! Please find our clarifications to your comments below.
> > >
> > > > “*the paper could be improved if you provided a screenshot of it*”
> > > - Thanks for the suggestion and we will include a screenshot in the final paper. Since we cannot include that in the response for now, we are providing further details here:
> > > 1. The questionnaires are conducted via Google Forms.
> > > 2. At the top of the form, we include the above mentioned instructions.
> > > 3. Immediately following the instructions are the question entries that we require workers to answer.
> > > 4. The question in each entry is simply the high-level task and the list of steps used to perform the task. The format is exactly the same as in Figure 1 and Figure 2 of the paper.
> > > 5. For each question, there are only two options “Yes” or “No”, from which workers must choose one.
> > >
> > > > “*a task like brewing coffee is open-ended, but there are certain necessary steps that you'd need to do to carry it out*”
> > > - Indeed, this is what we intend to measure with the *correctness* metric -- whether the action plans are sensible to humans for the corresponding tasks. We do not use this metric to measure whether the plan contains the a specific amount of details or whether it only contains allowed actions/objects, because these are judged by the *executability* metric. As we are measuring two axes simultaneously, the ultimate goal is to generate action plans that are both correct and executable.
> > > - As an example, a plan that looks like real actions is not necessarily actionable in an environment. For the following action plan that possibly could be generated by a raw language model:
> > > ```
> > > Task: Brew coffee
> > > Step 1: Use hot water and coffee beans to brew coffee
> > > ```
> > > According to our instructions, a worker may decide that the action plan is correct. However, it would fail at the executability metric and would not be a desirable action plan. In other words, we do not burden humans with the evaluation of “grounding”, which we also discuss in details below.
> > >
> > >
> > >
> > > > *“Possibly, giving the workers some notion of a 'ground truth' trajectory (aka the target amount of detail that you're looking for) would help address this issue.”*
> > >
> > > - While we agree that giving some notion of ‘ground truth’ can be useful, it is exceptionally difficult to train human annotators to determine an action plan is simultaneously *correct* and *executable* in the environment. This is because it is not practical to fully specify/enumerate all environment constraints, i.e. the set of allowed actions/objects and the commonsense constraints (e.g. “grab milk from fridge” before “open fridge”). Therefore, we choose to only ask human annotators to evaluate the “correctness” aspect of the plans and let the environment decide if they contain sufficient amount of detail to be “grounded” or “executable”.
> > >
> > > - Moreover, because of the lack of expressivity of VirtualHome, we do not want to bias human annotators towards action plans that could potentially be incomplete. An example for this is the ground-truth plan for the task “vacuum”:
> > > ```
> > > Task: Vacuum
> > > Step 1: Walk to home office
> > > Step 2: Walk to computer
> > > Step 3: Find vacuum cleaner
> > > Step 4: Switch on vacuum cleaner
> > > Step 5: Switch off vacuum cleaner
> > > ```
> > >
> > > - Furthermore, by evaluating only how sensible these generated action plans are, we can show the surprising finding that existing large language models already contain rich actionable knowledge. The problem that remains is *how to make them executable?*, which we show some initial efforts in this work and encourage future works to bridge the gap.

---

> > > > ### Comment · Reviewer_gxEx · 2021-11-30
> > > > **The evaluation is still not very convicing**
> > > >
> > > > Thanks! I understood all that from earlier. The issue, echoing what KhzN wrote, is that it is not clear that one can accurately decompose the task into two axes in this way. This is supported empirically by your own results, since machines score better than humans. I had suggested a way to improve it, which you could presumably try in revision... but oh well...
> > > >
> > > > Anyways, I vote to keep my new score (6).

---

> > > > > ### Author Response · Authors · 2021-11-30
> > > > > **Response to Reviewer gxEx**
> > > > >
> > > > > Thanks for the reply and we appreciate the further clarification of your concern!
> > > > >
> > > > > - Regarding your suggestion about human evaluations, we will take it into account (or a variant of it) in revision when we perform another round of human evaluations.
> > > > >
> > > > > - While we understand your concern and hope our earlier responses can be helpful, we would also like to clarify that when looking at two axes simultaneously, none of the language models or our proposed approach can match humans, i.e. the ground-truth has 100% executability and 55% correctness. We apologize for any confusion which may be caused by Figure 1.

---

### Official Review · Reviewer_mJtc · 2021-11-01

**Correctness:** 3
**Technical Novelty And Significance:** 2
**Empirical Novelty And Significance:** 2
**Recommendation:** 5
**Confidence:** 4

**Main Review:**

After rebuttal: I have read the author response. Some of my comments regarding the dataset and evaluation were partially addressed, so I am raising my score. However, I feel ambivalent about the paper due to concerns about the evaluation (See detailed comments in discussion thread).

====

Pros
* Using language models to infer actionable plans from text instructions is interesting
* Paper is generally easy to follow

Cons
* The paper doesn’t provide sufficient details about the data, the train and test splits, and the challenging aspects of the task.
* The pipelined approach considered here seems fairly limited
* Limited technical novelty
* Weak experiments

The paper should provide more details about the data. For instance, how similar are the train and test tasks? I would encourage the authors to provide a summary of the tasks. How was the 204/88 task split decided?

The metrics considered in the paper are less ideal and do not provide a holistic view of model performance. For instance,
* Executability doesn’t consider whether predicted actions are relevant to the task
* LCS penalizes actions that don’t appear in ground truth
* Correctness: Why is the correctness of GT plans only 55%? This makes it very hard to interpret these results and I am not sure how meaningful the results are.
Furthermore, it’s hard to say if the Translated variants in Table 1 are better than the plain language models. While these models are better in terms of executability and LCS, they are inferior in terms of correctness.

How were the human annotators instructed to label model generated plans?

Is it possible to define a straightforward metric that identifies whether a given task was successfully completed or not?

The proposed approach should be compared against other baselines. For instance, a fine-tuning baseline can be considered where the model translates instructions to plans.

I would encourage the authors to analyze and report model prediction errors.

Links to appendix are broken.


**Summary Of The Paper:**

This paper explores the ability of pre-trained language models to generate plans or action sequences from a text instruction. The in-context learning ability of language models is used where the model is prompted with an example instruction and corresponding action sequence and the query instruction. Since text sequences generated by the language model may not be directly usable in the agent environment, the closest valid text actions are identified using a retrieval approach. Experiments compare performance of different language models on tasks from the VirtualHome benchmark.

**Summary Of The Review:**

While the authors explore an interesting approach to planning based on language models, the paper is weak in terms of technical novelty and experiments.

---

> ### Author Response · Authors · 2021-11-21
> **Response to Reviewer mJtc (Part 1)**
>
> Thank you for the constructive feedback! We hope that we have addressed all your concerns below. We have also updated the paper based on your suggestions. If there are any further concerns that prevent you from accepting the paper, please let us know and we will attempt to address them!
>
> > *“The proposed approach should be compared against a fine-tuning baseline"*
>
> - As suggested, we added two fine-tuning baselines below. Note that these models are too big and if we fine-tune them locally, we cannot get good performance due to the engineering complexity and lack of large infrastructure. So, to be fair to these methods, we also trained a large model using OpenAI Fine-tuning API which has well-optimized training loops and infrastructure. With that, we are able to get good performance. However, our non-fine-tuned version still outperforms the carefully fine-tuned models in terms of executability while being worse in terms of LCS. This makes sense because the fine-tuned models can leverage large amounts of in-distribution data (~2000 pairs) to bias its generation, while our approach relies only on one example given at test time. Moreover, we note that compared to unsupervised approaches, there are several significant disadvantages for fine-tuning: 1) it is memory intensive and requires laborious engineering for large models, 2) it requires a considerable number of training examples, 3) it is biased by the coverage of a specific dataset and thus expected to perform more poorly on out-of-distribution data (as observed in [2]), and 4) it still requires large base model sizes to work well in the tasks we considered in this paper. See table below:
>
> | Models | Executability | LCS |
> | ------------- | ------------- | ------------- |
> | GPT-2 1.5B - finetuned locally | 4.71% | 23.71% |
> | GPT-3 12B - finetuned using OpenAI API | 66.07% | 34.08% |
> | **Ours - non-finetuned Codex 12B** | 78.57 % | 24.72 % |
>
>
> > *“The metrics considered in the paper are less ideal and do not provide a holistic view of model performance… Is it possible to define a straightforward metric that identifies whether a given task was successfully completed or not?"*
>
> - Due to the open-ended nature of the tasks we investigate in this paper, to the best of our knowledge, there isn’t a single straightforward metric that one can computationally produce. To compensate for this, we not only follow the VirtualHome paper to report executability and LCS [1], but we also measure the correctness of the action plans by conducting human studies. Therefore, the evaluations in this paper are mainly along two axes -- *executability* and *correctness* -- where executability measures grounding and correctness measures if an action plan achieves the desired goal. We are glad that Reviewer 28Tw finds the two axes “better characterize the strengths and limitations of the language models studied in this paper”. We also provide further analysis on the combined metric, i.e. action plans that are executable and correct, in Section 5.3. We do not focus on this combined metric because 1) entangling both axes would make it hard to show which axis to work on given that this is an early work in this direction, 2) it fails to demonstrate existing LLMs such as GPT-3 already contain rich actionable knowledge, and 3) it involves human studies so it is not scalable and cannot be computationally produced. Having said that, we also look forward to future works that investigate the evaluation aspect of these tasks.
>
>
> > *“Why is the correctness of GT plans only 55%? This makes it very hard to interpret these results and I am not sure how meaningful the results are."*
>
> - We would like to note that correctness is independently evaluated without environment visualization/execution. Thus it can even be considered as a separate benchmark and hopefully can provide insights into these results. Even though they are evaluated independently of the simulation environment, GT action plans, as well as those by Translated LMs, can only use allowed actions in the environment. Therefore, they are constrained by the expressivity of the environment. On the other hand, plain language models can generate action plans expressed in free-form language, which makes them possible to have even higher correctness than GT, although these plans are often not executable. Due to the open-ended nature and complexity of these tasks, it is also very challenging to simulate these tasks perfectly as judged by humans. Puig et. at., who collected the dataset, also report the correctness of these GT plans to be 64% in the VirtualHome paper [1], which is similar to our evaluation.

---

> > ### Comment · Reviewer_mJtc · 2021-11-29
> > **Evaluation needs to be more convincing**
> >
> > I thank the authors for the detailed response.
> >
> > Thanks for including the fine-tuning baseline. The fine-tuned GPT2 performance seems quite low - Have you looked into why this may be the case? If anything, I would assume that it would have good executability since it is directly trained on the kind of plans the environment expects?
> >
> > While I do think this work explores an important aspect of knowledge contained in language models, I still feel that the evaluation needs to be more convincing. While I agree with the authors on the difficulty of evaluating models in this setting, more analysis or experiments on other benchmarks can make the paper more compelling. For instance
> > * The correctness of ground truth plans is only 55% - Some analysis is necessary to understand why this is the case and if there are any systematic errors made by human judges.
> > * Can the approach be evaluated on a different benchmark where the plans can be directly executed against the environment so that we get a holistic view of executability and correctness?

---

> > > ### Author Response · Authors · 2021-11-30
> > > **Response to Reviewer mJtc**
> > >
> > > Thanks for the reply! Please find our clarifications to your comments below.
> > >
> > > > *”The fine-tuned GPT2 performance seems quite low - Have you looked into why this may be the case?”*
> > >
> > > - Qualitatively, inspecting the generated action plans by GPT-2, we find that there are several sources of errors:
> > > 1. It tends to generate longer sequences that contain unnecessary/irrelevant action steps because the model seems to (incorrectly) interpolate action plans from its training data.
> > > 2. As a result, it often violates the commonsense constraints of the environment (i.e. pre/post-conditions of actions).
> > > 3. The same issue of referring to unrecognizable actions/objects from the un-finetuned version still persists, but it occurs less often because the model has seen in-distribution data.
> > > - We hypothesize that this could be due to both an insufficient amount of fine-tuning data and the limited prior knowledge in the GPT-2 model. It could also be due to non-optimal training loops, so we also included another fine-tuned GPT-3 baseline and hope it could be a strong baseline.
> > >
> > > > *”Some analysis is necessary to understand why this is the case and if there are any systematic errors made by human judges”*
> > >
> > > - We would like to note that Puig et. at., who collected the dataset, also reported the correctness of these GT plans to be 64% in the VirtualHome paper. We independently ask a different group of human judges to evaluate the same dataset and obtain similar results. This validates that there is likely no systematic error during the process of evaluating the correctness by human judges.
> > > - We also do not observe systematic errors in the dataset itself (an uncurated subset is also provided in the Appendix).
> > > - The only source of the low executability is likely the lack of expressivity of the environment, i.e. certain tasks (e.g. “wash face”, “make popcorn”) cannot be fully achieved due to unimplemented actions/objects.
> > > - Although a brief analysis for this is provided in Section 4.3 of the paper, we agree that a more in-depth analysis including specific examples would be helpful and will include it in the final paper.
> > >
> > > > *”Can the approach be evaluated on a different benchmark where the plans can be directly executed against the environment so that we get a holistic view of executability and correctness?”*
> > >
> > > - Due to the open-ended nature of these tasks, to the best of our knowledge, there unfortunately isn’t another benchmark suitable for investigating this aspect of pre-trained language models.
> > > - While we understand readers may be concerned about the low correctness of the GT plans (due to limited expressivity of the environment), we hope the human evaluations could be a strong benchmark here because it is evaluated independently of any environment.
> > > - Having said that, we also look forward to future works that investigate the evaluation aspect of these tasks.

---

> ### Author Response · Authors · 2021-11-21
> **Response to Reviewer mJtc (Part 2)**
>
> > *“it’s hard to say if the Translated variants in Table 1 are better than the plain language models. While these models are better in terms of executability and LCS, they are inferior in terms of correctness."*
>
> - We acknowledge that the proposed approach is a trade-off rather than one best solution. The goal of this work is not to propose a technically novel solution that outperforms existing baselines in all the metrics. Instead, we aim to investigate what has been previously under-explored -- is there actionable knowledge in large language models and how can we extract it to use for embodied agents? To this end, we propose a simple viable approach that demonstrates such possibility, and we hope/encourage future efforts to build upon this work to bridge the current gap.
>
> > *“The paper doesn’t provide sufficient details about the data, the train and test splits, and the challenging aspects of the task."*
>
> - As suggested, we have added more details about the dataset and the train/test splits in the Appendix. Specifically, the tasks come from the dataset collected by the VirtualHome paper [1], and the training and testing tasks were selected randomly and kept fixed throughout this work. These tasks are particularly challenging for current AI systems because they are open-ended and expressed in free-form language. Most notably, these tasks represent the complex activities that humans perform every day. Expressed in free-form language, they can also be arbitrarily complex and with arbitrary variations, e.g. an agent can be commanded to `complete surveys on Amazon Turk` and `turn on TV with remote control` (in contrast to just `turn on TV`).
>
> > *“How were the human annotators instructed to label model generated plans?"*
>
> - As suggested, we have updated the paper with additional details about human evaluations in the Appendix. A briefer version is provided below.
> - Human evaluations are conducted on Amazon Mechanical Turk. For each method, we generate action plans for all 88 high-level tasks. The evaluations are conducted in the form of questionnaires containing all action plans with *unknown corresponding methods* and with *randomly permuted order*. The questionnaire contains the following instructions at the top:
>
> ``
> For every question below, determine whether the task can be completed in any reasonable scenario using the provided steps. In other words, can the task be decomposed into these steps? Note that simply re-stating the task does not mean completing it.
> ``
>
> - Human annotators are required to provide “Yes” or “No” to all the questions in the questionnaire, indicating if they believe the action plan completes the task. For each method, we report the correctness percentage averaged across 10 participating human annotators.
>
> > *“Links to appendix are broken."*
>
> - Thanks for pointing it out! We have fixed the links now.
>
>
> #### References
>
> [1] VirtualHome: Simulating HouseHold Activities via Programs. Xavier Puig, Kevin Ra, Marko Boben, Jiaman Li, Tingwu Wang, Sanja Fidler, and Antonio Torralba. CVPR2018.
>
> [2] Commonsense Knowledge Mining from Pretrained Models. Joe Davison, Joshua Feldman, Alexander Rush. ACL2019.

---

### Official Review · Reviewer_28Tw · 2021-11-01

**Correctness:** 3
**Technical Novelty And Significance:** 3
**Empirical Novelty And Significance:** 4
**Recommendation:** 8
**Confidence:** 5

**Main Review:**

## What I like about this paper

- Rather than trying to solve a particular task using large language models, this paper focuses on analyzing what is the capability of pre-trained language models in terms of decomposing high-level tasks into a sequence of instructions. In other words, what kind of actionable knowledge, if any, is stored in those large language models?

- Using two axes of evaluation - executability, and correctness - better characterize the strengths and limitations of the language models studied in this paper. Looking at the discrepancy between the correctness, as reported by the human annotators, and the executability of a plan motivates the need for a translation LM. It can overcome some of the restrictions interactive environments have, i.e. their action space contains instructions with a specific syntax.

- I like the suggestion of using another language to translate generated instructions to grounded actions to boost executability. Also, it makes sense to me to use an autoregressive trajectory correction approach to improve the executability of a plan.

## Concerns

- My main concern lies with the type of language models used in this analysis. Other than being really popular, it is not clear why the authors decided to only investigate models in the GPT-* family? Is it that other large language models (e.g., T5) are less amenable to prompt-engineering? To me, the main message/claims of the paper seem to be about large language models in general. It would be interesting to know if actionable knowledge is only present in GPT-* models or not.


-----
### Typos
- p.7: "We find that despite [being] more similar ..." - Missing word.
- p.7: "...; we find that for [a] many tasks we ..." - Out-of-place word.
- p.7: "... by [our -> the] human annotators" - Suggested change.

**Summary Of The Paper:**

This paper investigates whether large language models (LLMs) are capable, without additional training, of decomposing high-level tasks into a sequence of instructions (i.e., a plan) and grounding them in an embodied environment. Specifically, they relied on prompt engineering to provide enough context to the LLMs for generating sensible instructions. The authors also suggest using another language model to translate generated instructions into actions that are parsable by the embodied environment.

Through a series of experiments, the authors empirically show that larger LLMs are able to produce sensible plans (according to human annotators) that can also be mapped to executable actions within an embodied environment. In their experiments, the authors evaluate GPT-2, GPT-3, and Codex (including different sizes for the GPT-* family). Part of the paper is also dedicated to answering several hypotheses related to the investigation.

**Summary Of The Review:**

In addition to their impressive performance on several standard NLP tasks, large language models have been found useful for a wide variety of tasks when prompted appropriately. I believe understanding, via investigative work, what is the extent of the knowledge contained in those models to be important as it can better guide the research community towards what to explore next. The proposed paper is exactly about doing such an investigation. I found the paper well-motivated, and it includes the experiments to support the authors' analysis and discussion. For those reasons, I recommend this paper be accepted at ICLR.

---

> ### Author Response · Authors · 2021-11-21
> **Response to Reviewer 28Tw**
>
> Thank you for the constructive feedback! We have corrected the typos pointed out in the review. Please find below for answers to your comments.
>
>
> > *“Other than being really popular, it is not clear why the authors decided to only investigate models in the GPT-\* family? Is it that other large language models (e.g., T5) are less amenable to prompt-engineering?"*
>
> - Indeed. A major reason we focus our investigations on GPT-* family is that they can perform in-context learning from the prompt and do not require any gradient updates. This makes them have several major benefits over masked language models for our investigations: 1) they only require as few as one single “training” example provided as part of the prompt, so they can easily be adapted to new environments and domains, 2) they do not require fine-tuning to be used for action planning, so we can access larger sizes of them with limited GPU memory budget, and 3) they fit within existing model serving pipelines better, so we can easily access these large models through an inference-only API (such as the OpenAI API and the Hugging Face API used in this work).
> - That being said, we believe that the same actionable knowledge also exists in other large language models, such as T5, and it would be useful future work to study if they would be superior to the GPT-* family.

---

> > ### Comment · Reviewer_28Tw · 2021-11-23
> > **Thanks for the response**
> >
> > That makes sense. I agree that would be interesting to investigate as future work.
> >
> > Regarding the response to Reviewer VpdM about
> > > However, both Room2Room and ALFRED require visual input. Moreover, ALFRED requires object mask prediction, which likely requires environment-specific training data and carefully-designed architectures.
> >
> > Maybe the [ALFWorld](https://alfworld.github.io/) framework could be used? It builds on ALFRED and it seems to alleviate the needs for visual inputs and masks (i.e., there is a text version of ALFRED's environments).

---

> > > ### Author Response · Authors · 2021-11-30
> > > **Response to Reviewer 28Tw**
> > >
> > > Thanks for the reply! Please find our clarifications to your comments below.
> > >
> > > > *”Maybe the ALFWorld framework could be used?”*
> > >
> > > Thanks for the pointer! We agree that it would be applicable and interesting to evaluate our method in ALFWorld, but for a purpose complementary to the main contribution of this work, which we discuss in detail below.
> > >
> > > - While we focus on how well LLMs can decompose open-ended human activities (e.g. make breakfast, make coffee, etc) into sequences of actionable steps, ALFWorld considers only 6 task types (e.g. Pick & Place, Heat & Place, Cool & Place, etc). To a certain extent, our investigation abstracts away these mid-level tasks by omitting their detailed execution and focuses on higher-level tasks expressed in arbitrary free-form language.
> > >
> > > - Moreover, when evaluated without visual observation in ALFWorld, a frequent procedure is repeatedly searching for an object from many receptacles like drawers or cabinets based on environment feedback. For example, to find a vase, a text-only agent may need to repeatedly go to many drawers/cabinets one by one, open them, and check if a vase is present. If not, the agent must similarly check another drawer/cabinet before proceeding to the next step. To this end, a major challenge posed by ALFWorld is identifying a specific instance of an object by relying on environment feedback. In contrast, we simplify this problem by assuming only one instance of the same object is present and focus our studies on whether the action plans contain all commonsense steps (e.g. “open drawer” before “grab vase from drawer”) by relying on VirtualHome’s mechanism to correctly identify corresponding objects.
> > >
> > > - Nevertheless, although we don’t investigate ALFWorld in this work, ALFWorld provides a useful text-only framework to incorporate environment feedback, which we believe would be an exciting future direction.

---

> > > > ### Comment · Reviewer_28Tw · 2021-11-30
> > > > **Thanks for the response**
> > > >
> > > > That makes without providing context (current observation + history) that would be hard to find particular objects. I look forward to this future direction.

---

### Official Review · Reviewer_KhzN · 2021-11-02

**Correctness:** 3
**Technical Novelty And Significance:** 2
**Empirical Novelty And Significance:** 2
**Recommendation:** 3
**Confidence:** 4

**Main Review:**

This paper shows that the pretrained language model contains knowledge that is useful for action planning. While this is a promising direction to use pretrained language model, I have several concerns and questions for this paper.

1. Grounding & planning vs. knowledge extraction. When we do grounding or planning, the agent’s actions may change given the environment they are put in. The proposed approach is closer to procedure knowledge extraction using language models rather than planning or grounding. While the available actions in VirtualHome might be different from other virtual environments (therefore, this paper proposed a translation step), the generated action plans don’t change if we change the environment from one home to another. So, the generated plans are not really grounded in the domain or environment the agent is planning for. The authors may consider repositioning this paper to knowledge extraction papers that fit the contribution better.

2. Choice of VirutalHome task for planning. The tasks are mainly around the daily home tasks like setting up a dinner table. To perform such tasks, most of the time, the agent may just apply predefined procedure knowledge. However, procedure knowledge and the translation approach proposed by this paper cannot plan for the task like “stack two plates on the right of a cup” because the language model doesn’t ground its knowledge to the environment so the agent cannot tell the difference between putting down on the left or right of the cup. This is the main limitation to using the proposed approach as a planner.

3. Evaluating correctness and executability separately. The goal of task planning is to find the plan that is executable and correct, i.e. success rate. However, most of the tables in the evaluation only show one or another. Only Table 4 shows “# of C and E”. If we compute the success rate for Translated Codex 12B, it is 15/88=0.17 and would be lower for other models. This reflects the performance of the proposed method better and the overall success rate is quite low.

4. Missing related work. This paper missed some related work on knowledge extraction/mining using pertained language models. Just to list a few below. The authors will need to discuss how the proposed approach is different from other knowledge mining approaches compared to prior work.
 - Petroni, F., Rocktäschel, T., Lewis, P., Bakhtin, A., Wu, Y., Miller, A.H. and Riedel, S. “Language models as knowledge bases?” In EMNLP 2019.
 - Davison, Joe, Joshua Feldman, and Alexander M. Rush. "Commonsense knowledge mining from pretrained models." In EMNLP 2019.
 - Jiang, Z., Xu, F.F., Araki, J. and Neubig, G. “How can we know what language models know?”. Transactions of the Association for Computational Linguistics, 8, pp.423-438.


**Summary Of The Paper:**

This paper proposes using pretrained language models for planning actions. They achieve this by providing an example task sequence as a prompt to generate an action description for each step autoregressively. At each step, they translate the action description to the environment action by computing the similarity between the description and the available environment actions. The results show that there is a trade-off between executability and correctness.

**Summary Of The Review:**

This paper shows promising results on extracting executable actions using pretrained language model. However, there are major concerns about how this relates and contributes to planning and grounding.

---

> ### Author Response · Authors · 2021-11-21
> **Response to Reviewer KhzN (Part 1)**
>
> Thank you for the constructive feedback! Please find below for answers and clarifications to your comments. Based on your suggestions, we have also updated the paper (e.g. related works, experiments and other sections). If there are any further concerns that prevent you from accepting the paper, please let us know and we will attempt to address them!
>
> > *“the generated plans are not really grounded in the domain or environment the agent is planning for.”*
>
> - Our goal is to study if the knowledge already contained in the language models can be extracted to achieve high-level tasks for embodied agents given access to low-level controllers, *without fine-tuning on environment-specific data*.
> - Therefore, instead of strongly grounding the LLM generation by using downstream data from the environment, we focused on grounding with weak constraints such that we evaluate LLMs raw knowledge as closely as possible.
> - To generate action plans for complex human activities such as `make the bed` and `make coffee`,  we’d like the weak constraints to ensure these plans obey common-sense knowledge, which is measured by the environment pre/post-conditions. By satisfying these pre/post-conditions, an action plan can indeed be executed in an embodied environment such as VirtualHome. For an atomic action `open`, one pre-condition is that the agent cannot grab milk from the fridge before opening it, and one post-condition is that the state of the fridge changes from “closed” to “open”.
>
> > *“The authors may consider repositioning this paper to knowledge extraction papers that fit the contribution better."*
>
> - While we agree that knowledge extraction is relevant to this work, we believe there are three fundamental differences between this work and knowledge extraction papers: 1) While the common goal in knowledge extraction is obtaining knowledge for people to use, our goal here is to use this knowledge directly for autonomous agents to act in interactive environments, 2) As a result, the proposed approach needs to care about what actions are allowed and what objects are available to use, whereas knowledge extraction does not have such constraint, 3) The LM output needs to satisfy all the pre/post-conditions of the sequentially executed actions in an interactive environment like VirtualHome, whereas knowledge extraction is not concerned with pre/post-conditions.
>
>
> > *“The tasks are mainly around the daily home tasks like setting up a dinner table. To perform such tasks, most of the time, the agent may just apply predefined procedure knowledge. However, procedure knowledge and the translation approach proposed by this paper cannot plan for the task like “stack two plates on the right of a cup"*
>
> - Since the proposed approach does not train on VirtualHome-specific data, our qualitative observations show that it can in fact generalize well beyond the home setting. Two uncurated examples are `eat at a restaurant` and `do workout at a gym`, which we show below. Unfortunately, to the best of our knowledge, there isn't an environment capable of executing these tasks in the same fashion as VirtualHome, so we do not investigate them in the paper.
>
> ```
> Task: Eat at a restaurant
> Step 1: Walk to restaurant
> Step 2: Walk to table
> Step 3: Find chair
> Step 4: Sit on chair
> Step 5: Find menu
> Step 6: Look at menu
> Step 7: Read menu
> Step 8: Order food and drink from waiter/waitress
> Step 9: Eat food and drink ordered
>
> Task: Do workout at a gym
> Step 1: Walk to gym
> Step 2: Walk to locker
> Step 3: Find locker
> Step 4: Unlock locker
> Step 5: Remove clothes from locker
> Step 6: Put on workout clothes
> Step 7: Find workout machine
> Step 8: Sit on workout machine
> Step 9: Start working out
> ```
>
> - Furthermore, since the tasks are expressed in free-form language, they can be arbitrarily complex or with arbitrary variations; we observe that not only LLMs can achieve common tasks like `setting up a dinner table`, but they contain actionable knowledge for tasks with specific requirements, such as the proposed task `stack two plates on the right of a cup` which we also show below. Although we cannot apply the proposed translation procedure due to the lack of expressivity of the VirtualHome environment, we believe this work provides a well-founded starting point for future works to build upon.
>
> ```
> Task: Stack two plates on the right of a cup
> Step 1: Walk to kitchen
> Step 2: Get a cup
> Step 3: Walk to right side of table
> Step 4: Pick up the first plate
> Step 5: Put it on the right side of the cup
> Step 6: Pick up the second plate
> Step 7: Put it on the right side of the cup
> ```
>
> Note: The above action plans are generated by `GPT-3 175B` with the same procedure outlined in Section 3.1.

---

> > ### Comment · Reviewer_KhzN · 2021-11-28
> > **Thanks for the response!**
> >
> > I thank the authors for the response and the updates in the paper!
> >
> > I agree that characterizing the language model in two axes can help understand the limitation but it is also important to show the percentage of the executable and correct plans so the readers can understand the performance gap.
> >
> > The additional examples are helpful to show the proposed method can extend to non-VirtualHome data. However, the authors still need to clarify the "grounding" part. Grounding usually means that the agent changes its plan based on the environment or the state of the world. For example, when the milk is not in the fridge but in a shopping bag, "open the fridge" is no longer a pre-condition. If the agent doesn't change its plan when we swap the environment or state of the world, it does not really perform grounding. I would like to see how the proposed method adapts its plans based on the environments, or the authors may need to avoid using "grounding" in the paper.

---

> > > ### Author Response · Authors · 2021-11-30
> > > **Response to Reviewer KhzN**
> > >
> > > Thanks for the reply! Please find our clarifications to your comments below.
> > >
> > > > *”it is also important to show the percentage of the executable and correct plans so the readers can understand the performance gap.”*
> > >
> > > - We agree that showing the percentage of the executable and correct plans is important to show the performance gap between the current method and the ground-truth, and we will make sure to include that in the final version of the paper.
> > >
> > > > *”the authors still need to clarify the "grounding" part”*
> > >
> > > - We would like to clarify that we loosely refer “grounding” as “the ability to generate *executable* action plans in an embodied environment”. Indeed, one limitation of our approach is that we do not condition on environment state, so the proposed method will not generate different plans based on new environment state, unless relevant information is appropriately incorporated in the prompt (e.g. “grab milk from the fridge” or “grab milk from shopping bag”).
> > >
> > > - However, we would like to note that incorporating state information or adapting to new state both subject to expressivity of a specific environment and likely require considerable amount of *training data*. We thus do not investigate it in this work because we would like to focus our investigation on only non-finetuned pre-trained language models, as they can provide the most useful indicators whether pure language pre-training can internalize *general-purpose* and *actionable* knowledge.
> > >
> > > - Having said that, we understand that the word “grounding” could lead to some confusion for the readers, so we will modify the paper accordingly to clearly express the contributions of this work.

---

> ### Author Response · Authors · 2021-11-21
> **Response to Reviewer KhzN (Part 2)**
>
> > *“Correctness and executability are evaluated separately."*
> - We agree that a correct and executable action plan is a more direct measurement of successes. However, same as Reviewer 28Tw, we believe that evaluating two axes separately can “better characterize the strengths and limitations of the language models studied in this paper”, which can guide future efforts to determine what aspects to work on. In addition, entangling the two metrics fails to demonstrate existing LLMs such as GPT-3 already contain rich actionable knowledge, and the combined metric would involve human studies so it is not scalable and cannot be computationally produced.
> - Moreover, due to the open-ended nature of these tasks, evaluating success rate is remarkably challenging. To provide insights into the actionable knowledge in LLMs as comprehensively as possible, we not only follow the VirtualHome paper to report executability and LCS [1], but we also additionally conduct human evaluations that can be considered as a separate benchmark.  Having said that, we also look forward to future works that investigate the evaluation aspect of these tasks.
>
>
> > *“[Simultaneously being executable AND correct] reflects the performance of the proposed method better and the overall success rate is quite low."*
>
> - We would like to note that since annotated ground-truth have human-evaluated correctness of 55%, this metric (i.e. the “success rate”) is compounded with the expressivity of the environment and has a maximum of 55%. Although we agree that there is still a long way to obtain near-perfect action plans for arbitrary open-ended tasks, the proposed approach improves the current baseline by a large margin and we hope it serves as a foundation for future works.
>
>
> #### References
>
> [1] VirtualHome: Simulating HouseHold Activities via Programs. Xavier Puig, Kevin Ra, Marko Boben, Jiaman Li, Tingwu Wang, Sanja Fidler, and Antonio Torralba. CVPR2018.

---

### Official Review · Reviewer_VpdM · 2021-11-03

**Correctness:** 3
**Technical Novelty And Significance:** 2
**Empirical Novelty And Significance:** 3
**Recommendation:** 5
**Confidence:** 4

**Details Of Ethics Concerns:**

In this work, human evaluation is the major experimental evaluation metric, but it is unclear how it is done and whether the process is legal and fair. For example, how are the workers paid? Do the authors have a human subject approval for conducting the experiments?

**Main Review:**

Strengths:
- LLMs are powerful tools and learn knowledge that we may even not know. So prompting them to utilize the learned knowledge is a promising research direction. Prompting LLMs for actionable knowledge extraction is new and interesting.
- Different LLMs including GPT-3 and Codex are used and evaluated here.
- The prompt and translation method is reasonable on the VirtualHome environment.

Weaknesses:
- The paper's claim is to extract actionable knowledge for embodied agents, but in fact, it seems the translation process is particularly tailored for VirtualHome. I couldn't see a clear way to generalize to other embodied environments. It would be great if the authors can show its generalization to other environments such as Room2Room and ALFRED. I am not sure if the translation part is really necessary in other environments.
- Moreover, it seems there is no embodied agent to execute those instructions. All experiments are conducted on the text level only, but we all know that there is a huge gap between text instructions and actual actions taken by the agent. So it would be much more convincing if the authors can show an embodied agent can actually follow the generated instructions and complete tasks in the environment.
- There seems to be a tradeoff between executability and correctness for the current approach. The correctness is significantly dropped from 56% to 34% for Translated Codex 12B. Why are some correctness numbers missing in Table 1? What is the correctness of Translated GPT-3 175B? The Correctness metric is also missing in Table 2 and Table 3. Why?
- It is actually pretty easy to improve the executability by finding the most similar actions from the action set of the environment. The real challenge is to make the actions more executable while preserving their correctness, which, however, isn't achieved in this work and it quite a pity. Without considering the correctness, executability does not really matter much, for an extreme example,  a random agent may achieve  high executability but low correctness.
- Many technical details are missing in the paper, for example,
  - It is unclear how Executability is calculated. There is only high-level language description in the paper. Can you elaborate it?
  - It is unclear how the sampling for LMs is done. Do you sample multiple ones and then rely on human evaluation to select the best one? This doesn't sound right and scalable.
  - Section 2.1: Appendix ?? points to nowhere. Similar issues exist in many places of the paper.
  - "" in the paper is in the wrong format.


**Summary Of The Paper:**

This paper takes advantage of the pre-trained large language models (LLMs) and assesses whether they can be helpful for embodied agents. The authors choose the VirtualHome environment as the testbed and prompt the LLMs to decompose a high-level instruction to detailed, actionable instructions. Human evaluation is conducted to evaluate the correctness and executability of the generated instructions.

**Summary Of The Review:**

I like the idea of prompting LLMs to decompose high-level instructions and extract actionable knowledge, which is a novel perspective for embodied agents. However, the execution of the idea and the experiments are not right there to support the claim. It would be great if the authors can show its effectiveness on "actual" embodied agents in different environments instead of text games.

---

> ### Author Response · Authors · 2021-11-21
> **Response to Reviewer VpdM (Part 1)**
>
> Thank you for the constructive feedback! We hope that we have addressed all your concerns below. We have also updated the paper based on your suggestions on technical details. If there are any further concerns that prevent you from accepting the paper, please let us know and we will attempt to address them!
>
> > *“it seems the translation process is particularly tailored for VirtualHome. I couldn't see a clear way to generalize to other embodied environments. It would be great if the authors can show its generalization to other environments such as Room2Room and ALFRED."*
>
> - The translation process only requires the set of all available actions in an embodied environment. To generalize to other environments similar to VirtualHome or domains beyond the household setting, one may simply supply another set of available actions. Given that the proposed approach utilizes a pre-trained language model without additional training for the VirtualHome environment, it is likely that the approach can generalize to new environments better than other methods that require additional training, such as a fine-tuning baseline.
> - Having said that, there isn’t another environment suitable for our investigations to the best of our knowledge. We are interested in investigating raw actionable knowledge in LLMs *as is* and *without fine-tuning*. However, both Room2Room and ALFRED require visual input. Moreover, ALFRED requires object mask prediction, which likely requires environment-specific training data and carefully-designed architectures. Although our approach can be potentially adapted to these environments, we do not investigate it in this work as it conflates our contributions.
>
>  > *“it would be much more convincing if the authors can show an embodied agent can actually follow the generated instructions and complete tasks in the environment."*
>
> - The agent is indeed following generated instructions and completing the tasks (please find the video results of some visualized action plans in the [website](https://sites.google.com/view/language-model-as-planner)). In VirtualHome, low-level controllers are assumed given. Therefore, actions are mid-level (e.g. `walk to kitchen`, `open fridge`) and are expressed as text instructions with a specific format. To complete a high-level task (e.g. `make breakfast`), an embodied agent needs to execute a series of these mid-level actions (see Section 1 in Appendix for an example). In this work, we thus study whether large language models contain actionable knowledge for these high-level tasks *without further training*, and we propose an approach to extract executable action plans from LLMs to drive an embodied agent to complete these tasks. The entire process requires no additional training data and involves no human supervision.
>
> > *“There seems to be a tradeoff between executability and correctness for the current approach. The correctness is significantly dropped from 56% to 34% for Translated Codex 12B."*
>
> - Indeed. We are currently observing a tradeoff between executability and correctness. However, we would also like to note that correctness is evaluated without environment execution/visualization. Therefore, Vanilla LMs enjoy the benefit of being able to generate free-form text and are not concerned with generating actions that can be executed by an embodied agent (thus more likely to have higher correctness). On the other hand, human-annotated ground-truths and Translated LMs need to comply with the constraint that actions must be allowed by the environment. To this end, the drop in correctness is also partially attributed to the limited expressivity of the environment.
>
>
> > *“Why are some correctness numbers missing in Table 1? What is the correctness of Translated GPT-3 175B? The Correctness metric is also missing in Table 2 and Table 3."*
>
> - Due to the high cost of human evaluations, we originally only evaluated a representative subset of all the models. We had also mentioned this reason in the paper already in Footnote 4.
> - However, upon reviewer’s suggestion, we decided to perform another round of human evaluations and have added the missing numbers in Table 1 -- please see the updated draft, we have marked changes in red.
>
> > *“The real challenge is to make the actions more executable while preserving their correctness, which, however, isn't achieved in this work and it quite a pity."*
>
> - Generating executable actions while preserving correctness is a very challenging goal, especially for complex human activities/tasks considered in this work. Since this is a new direction that all reviewers find interesting and promising, we hope this is the first step towards this goal.

---

> ### Author Response · Authors · 2021-11-21
> **Response to Reviewer VpdM (Part 2)**
>
> > *“It is unclear how Executability is calculated. There is only high-level language description in the paper. Can you elaborate it?"*
>
> - Executability is a measure reported by the VirtualHome environment and has also been used in previous works [1, 2]. Specifically, for an action plan to be executable, it must 1) contain actions that can be recognized and parsed by the environment, 2) not violate any pre/post conditions of the actions. The pre/post conditions for each atomic action mostly represent commonsense knowledge and were developed as part of the VirtualHome simulator. For example, to open an “object”, the list of preconditions include 1) “object” must be openable and is currently closed, 2) the agent is close to “object”, 3) “object” is not in some closed container, and 4) the agent must have at least one free hand. A post-condition of opening an “object” is that the state of the “object” changes from closed to open. A fuller list of pre/post conditions of actions can be found in [the VirtualHome documentation](http://virtual-home.org/documentation/kb/actions.html).
>
>
> > *“It is unclear how the sampling for LMs is done. Do you sample multiple ones and then rely on human evaluation to select the best one? This doesn't sound right and scalable."*
>
> - We sample LMs by using nucleus sampling and temperature sampling. And we indeed sample multiple ones, and we use mean token log probability and action matching score to select the best sample. The entire sampling procedure does not involve any human labor. Furthermore, for all methods, we optimize the number of samples taken using hyperparameter sweep and report the best result for each individual method. More details can be found in Sections 3.1 and 3.2 of the paper.
>
>
> > *“In this work, human evaluation is the major experimental evaluation metric, but it is unclear how it is done”*
>
> - As suggested, we have updated the paper with additional details about human evaluations in the Appendix. A briefer version is provided below.
> - Human evaluations are conducted on Amazon Mechanical Turk. For each method, we generate action plans for all 88 high-level tasks. The evaluations are conducted in the form of questionnaires containing all action plans with *unknown corresponding methods* and with *randomly permuted order*. The questionnaire contains the following instructions at the top:
>
> ``
> For every question below, determine whether the task can be completed in any reasonable scenario using the provided steps. In other words, can the task be decomposed into these steps? Note that simply re-stating the task does not mean completing it.
> ``
>
> - Human annotators are required to provide “Yes” or “No” to all the questions in the questionnaire, indicating if they believe the action plan completes the task. For each method, we report the correctness percentage averaged across 10 participating human annotators.
>
> > *”how are the workers paid? Do the authors have a human subject approval for conducting the experiments?"*
>
> - We pay the workers an hourly rate of $15 US Dollars. The evaluation is done online via Amazon Mechanical Turk. The Turkers voluntarily answer the questions with the abovementioned hourly pay rate as compensation.
>
> #### References
> [1] VirtualHome: Simulating HouseHold Activities via Programs. Xavier Puig, Kevin Ra, Marko Boben, Jiaman Li, Tingwu Wang, Sanja Fidler, and Antonio Torralba. CVPR2018.
>
> [2] Synthesizing Environment-Aware Activities via Activity Sketches. Yuan-Hong Liao, Xavier Puig, Marko Boben, Antonio Torralba, and Sanja Fidler. CVPR2019.

---

### Author Response · Authors · 2021-11-21
**Summary of Updates and Responses to Common Questions (Part 1)**

We thank the reviewers for the insightful feedback. We’re excited that all reviewers unanimously agree that extracting actionable knowledge from LLMs is novel, interesting, and promising. We’re also glad that reviewers find that the paper is “easy to follow” (mJtc), “presents a simple idea that greatly improves performance” (gxEx), and that it can “guide the research community towards what to explore next” (28Tw, gxEx). We are pleased to report that we have completed the experiments suggested by the reviewers and report the common answers below. Full responses are in the individual replies to each reviewer. Updates to the manuscript are highlighted in red.

> *"The proposed approach should be compared against other baselines. For instance, a fine-tuning baseline can be considered where the model translates instructions to plans."*

- We have updated the manuscript with a fine-tuning baseline as suggested by Reviewer mJtc.  In summary, smaller models such as a fine-tuned GPT-2 1.5B perform poorly in all the metrics, and larger models such as a fine-tuned GPT-3 12B obtains higher LCS but still underperforms our unsupervised prompting approach in executability. Moreover, fine-tuning has several significant disadvantages compared to prompting, such as infrastructure/engineering complexity, requiring training data, and poorer performance on out-of-distribution tasks. See table below for comparisons:

| Models | Executability | LCS |
| ------------- | ------------- | ------------- |
| GPT-2 1.5B - finetuned locally | 4.71% | 23.71% |
| GPT-3 12B - finetuned using OpenAI API | 66.07% | 34.08% |
| **Ours - non-finetuned Translated Codex 12B** | 78.57 % | 24.72 % |

> *"Why are some correctness numbers missing in Table 1? What is the correctness of Translated GPT-3 175B?"*

- As suggested by Reviewer VpdM, we have updated the manuscript with additional human evaluations for the *correctness* measure on remaining models.  In summary, the evaluations are consistent with our previous findings: 1) large language models contain rich actionable knowledge which scales with its model sizes, and 2) by constraining the list of allowed actions an agent can take, we observe a drop in correctness but a much higher executability. New human evaluations on our **Translated GPT-3 175B** suggest that such trade-off can be bridged with increased model sizes. See table below for comparisons:

| Models | Executability | Correctness |
| ------------- | ------------- | ------------- |
| GPT-2 0.1B | 18.66% | 14.27% |
| GPT-2 1.5B | 39.40% | 19.51% |
| Codex 12B | 18.07% | 56.06% |
| GPT-3 175B | 7.79% | 65.19% |
| **Ours - Translated Codex 12B** | 78.57 % | 34.97 % |
| **Ours - Translated GPT-3 175B** | 73.05 % | 51.73 % |


> [Reviewer VPDM, KHZN, GXEX] *Concerns related to the grounding of action plans generated by LLMs.*

- Our goal is to study if the knowledge already contained in the language models can be extracted to achieve high-level tasks (e.g. `make breakfast`) for embodied agents given access to low-level controllers (e.g. `open fridge`). Although we added the fine-tuning baseline above on reviewers’ suggestion, we specifically focus on using the knowledge of LLMs *as is*, without any finetuning, as is often the case in evaluating downstream tasks in the NLP community.
- However, we find that it is not possible to directly execute free-form action plans generated by LLMs without any form of grounding. Instead of strongly grounding the LLM generation by using downstream data from the environment, we focus on grounding with weak constraints such that we evaluate LLMs raw knowledge as closely as possible.
- To generate action plans for complex human activities such as `make the bed` and `make coffee`,  we’d like the weak constraints to ensure these plans obey common-sense knowledge, which is measured by the environment pre/post-conditions and is encapsulated by the *executability* measure. For an action plan to be grounded (i.e. executed), it must satisfy all the actions’ pre-conditions (e.g. the agent cannot grab milk from the fridge before opening it) and post-conditions (e.g. the state of the fridge changes from “closed” to “open” after the agent opens it).
- To ground the output by LLMs, we propose to generate a single instruction at a time and translate it into an allowed action, entirely in an unsupervised manner. This not only ensures that each action can be performed by the agent, but by autoregressively conditioning on past actions, it allows the LLM to reason about the necessary future actions, including those that are part of the common-sense knowledge.

---

### Author Response · Authors · 2021-11-21
**Summary of Updates and Responses to Common Questions (Part 2)**

> [Reviewer VPDM, KHZN, MJTC, GXEX] *Is there a single metric that can provide a holistic view of model performance?*
- Due to the open-ended nature of the tasks we investigate in this paper, to the best of our knowledge, there isn’t a single straightforward metric that one can computationally produce. To compensate for this, we not only follow the VirtualHome paper to report executability and LCS [1], but we also measure the correctness of the action plans by conducting human studies. Therefore, the evaluations in this paper are mainly along two axes -- *executability* and *correctness* -- where executability measures grounding and correctness measures if an action plan achieves the desired goal. We are glad that Reviewer 28Tw finds the two axes “better characterize the strengths and limitations of the language models studied in this paper”. We also provide further analysis on the combined metric, i.e. action plans that are executable and correct, in Section 5.3. We do not focus on this combined metric because 1) entangling both axes would make it hard to show which axis to work on given that this is an early work in this direction, 2) it fails to demonstrate existing LLMs such as GPT-3 already contain rich actionable knowledge, and 3) it involves human studies so it is not scalable and cannot be computationally produced. Having said that, we also look forward to future works that investigate the evaluation aspect of these tasks.

> [Reviewer VPDM, MJTC, GXEX] *Additional details on human evaluations.*

- As suggested, we have updated the paper with additional details about human evaluations in the Appendix. A briefer version is provided below.
- Human evaluations are conducted on Amazon Mechanical Turk. For each method, we generate action plans for all 88 high-level tasks. The evaluations are conducted in the form of questionnaires containing all action plans with *unknown corresponding methods* and with *randomly permuted order*. The questionnaire contains the following instructions at the top:

``
For every question below, determine whether the task can be completed in any reasonable scenario using the provided steps. In other words, can the task be decomposed into these steps? Note that simply re-stating the task does not mean completing it.
``

- Human annotators are required to provide “Yes” or “No” to all the questions in the questionnaire, indicating if they believe the action plan completes the task. For each method, we report the correctness percentage averaged across 10 participating human annotators.


#### References

[1] VirtualHome: Simulating HouseHold Activities via Programs. Xavier Puig, Kevin Ra, Marko Boben, Jiaman Li, Tingwu Wang, Sanja Fidler, and Antonio Torralba. CVPR2018.

[2] ALFRED: A Benchmark for Interpreting Grounded Instructions for Everyday Tasks. Mohit Shridhar and Jesse Thomason and Daniel Gordon and Yonatan Bisk and Winson Han and Roozbeh Mottaghi and Luke Zettlemoyer and Dieter Fox. CVPR2020.

---

### Public Comment · ~Wenlong_Huang1 · 2022-02-05
**Newest Version of the Paper**

Please find the newest version of the paper at https://arxiv.org/abs/2201.07207

---

### Decision · Program_Chairs · 2022-01-20

**Decision:**

Reject

**Comment:**

This manuscript presents a method to refine high-level task descriptions into mid-level executable steps. The idea of using language models to generate steps for a robot to follow is very interesting. Reviewer concerns focused on the general applicability of the approach and the evaluation.

Reviewers pointed out that the method is tied to VirtualHome which has various properties that are in general not true: the action space is small, the action space is very sparse, and objects tend to be unique.

First, the method enumerates a sentence for every possible action and object combination in the environment. The fact that VirtualHome has few verbs and few objects and that neither of these has complex additional structure (adjectives, adverbs, etc.) means that this is practical. But in any other practical setting this will be impossible. The manuscript mentions this limitation and hints at possible ways to resolve it.

Second, the method requires that the action space must be incredibly sparse. Moreover, a set of common sense rules are needed which are environment specific and must be hand curated. VirtualHome disallows microwaving a cup for example. It also disallows opening the TV. Both of these are valid actions that happen all the time.

Third, the method requires that objects be unique. If multiple plates, vacuum cleaners, lotions, etc. existed and had to be manipulated, e.g., there is no mechanism to refer to any one plate consistently. The model could generate something like "the first plate" but how to actually execute such an action is far from clear.

This third issue is related to the problem of grounding. Normally, grounding means connecting an abstract concept to something concrete in the environment. All of the grounding that is performed here is by virtue of VirtualHome having unique objects in its environments and the actions not requiring multiple instances of the same object. This is not addressing the problem of grounding. Reviewers requested that grounding be removed from the manuscript. This would significantly enhance it, as the model is inherently incapable of grounding as the authors say: "Indeed, one limitation of our approach is that we do not condition on environment state"

Reviewers took issue with details of the evaluation, which are largely a consequence of the choice of VirtualHome. Sometimes this manifested as strange results like models outperforming humans in terms of correctness. As reviewers pointed out, this is worrisome.

Reviewers were also concerned about the title. It implies that language models are zero-shot planners, but this is not the case. They are instead able to decompose actions into mid-level steps. Reviewers suggested that it would be better to focus the title and tone of the manuscript on extracting task/subtask structures from language models.

The idea presented here, that language models can break tasks into subtasks is interesting. But the manuscript goes a step further and discusses embodied agents which to reviewers appeared to be a reach: there is no grounding and in no sense is the output of the language model any different if the agent is embodied. Even the most positive reviewers felt that discussing embodied agents is unhelpful: it would be better to focus on task/subtask structures. And indeed, this would be more general. All of the concerns that reviewers had around the evaluation would be alleviated by focusing on a language task instead. And the effect of a narrow space of actions, constraints on those actions, and multiple objects of the same class, could be evaluated and reported. Even if the authors had to collect such a corpus, given the difficulties they describe in evaluating on VirtualHome, this would be less of a burden. This could be a strong submission in the future.